# NEURAL SDEs MADE EASY:
# SDEs ARE INFINITE-DIMENSIONAL GANs

## ABSTRACT

Several authors have introduced *Neural Stochastic Differential Equations* (Neural SDEs), often involving complex theory with various limitations. Here, we aim to introduce a generic, user friendly approach to neural SDEs. Our central contribution is the observation that an SDE is a map from Wiener measure (Brownian motion) to a solution distribution, which may be sampled from, but which does not admit a straightforward notion of probability density – and that this is just the familiar formulation of a GAN. This produces a continuous-time generative model, arbitrary drift and diffusions are admissible, and in the infinite data limit any SDE may be learnt. After that, we construct a new scheme for sampling *and reconstructing* Brownian motion, with constant average-case time and memory costs, adapted to the access patterns of an SDE solver. Finally, we demonstrate that the adjoint SDE (used for backpropagation) may be constructed via rough path theory, without the previous theoretical complexity of two-sided filtrations.

## 1 INTRODUCTION

Neural differential equations are an elegant concept, bringing together the two dominant modelling paradigms of neural networks and differential equations. Indeed, since their introduction, Neural Ordinary Differential Equations (Chen et al., 2018) have prompted the creation of a wide variety of similarly-inspired models, for example based around controlled differential equations (Kidger et al., 2020b; Morrill et al., 2020), Lagrangians (Cranmer et al., 2020), higher-order ODEs (Massaroli et al., 2020; Norcliffe et al., 2020), and equilibrium points (Bai et al., 2019).

In particular, several authors have introduced *Neural Stochastic Differential Equations* (neural SDEs), such as Tzen & Raginsky (2019a); Li et al. (2020); Hodgkinson et al. (2020) among others.

### 1.1 RELATED WORK

We begin by discussing previous formulations, and applications, of Neural SDEs.

Tzen & Raginsky (2019a;b) obtain Neural SDEs as a continuous limit of deep latent Gaussian models. They train by optimising a variational bound, using forward-mode autodifferentiation. They consider only theoretical applications, for modelling distributions as the terminal value of an SDE.

Li et al. (2020) give arguably the closest analogue to the neural ODEs of Chen et al. (2018). They introduce neural SDEs via a subtle argument involving two-sided filtrations and backward Stratonovich integrals, but in doing so are able to introduce a backward-in-time adjoint equation, using only efficient-to-compute vector-Jacobian products. In applications, they use neural SDEs in a latent variable modelling framework, using the stochasticity to model Bayesian uncertainty.

Hodgkinson et al. (2020) introduce Neural SDEs via an elegant theoretical argument, as a limit of random ODEs. The limit is made meaningful via rough path theory. In applications, they use the limiting random ODEs, and treat stochasticity as a regulariser within a normalising flow. However, they remark that in this setting the optimal diffusion is zero. This is a recurring problem: Innes et al. (2019) also train neural SDEs for which the optimal diffusion is zero.

Rackauckas et al. (2020) treat neural SDEs in classical Feynman–Kac fashion, and like Hodgkinson et al. (2020); Tzen & Raginsky (2019a;b), optimise a loss on just the terminal value of the SDE.

Briol et al. (2020); Gierjatowicz et al. (2020) instead consider the more general case of using a neural SDE to model a time-varying quantity, for which the stochasticity in the system models the variability (specifically certain statistics) of time-varying data. Letting $\mu, \nu$ denote the learnt and true distributions, both train by minimising $|\mu(f) - \nu(f)|$ for functions of interest $f$ (such as derivative payoffs). This corresponds to training with a non-characteristic MMD (Gretton et al., 2013).

Several authors, such as Oganesyan et al. (2020); Hodgkinson et al. (2020); Liu et al. (2019), seek to use stochasticity as a way to enhance or regularise a neural ODE model.

Our approach is most similar to Li et al. (2020), in that we treat neural SDEs as learnt continuous-time model components of a differentiable computation graph, and Briol et al. (2020); Gierjatowicz et al. (2020), in that we use stochasticity to model distributions on path space. The resulting neural SDE is not a improvement to a similar neural ODE, but a standalone concept in its own right.

## 1.2 CONTRIBUTIONS

Our central contribution is the observation that the mathematical formulation of SDEs is directly comparable to the machine learning formulation of GANs. Using this connection, we show how it becomes straightforward to train neural SDEs as generative time series models. Arbitrary drift and diffusions are admissible, and in the infinite data limit any SDE may be learnt.

Next, we introduce a new way of sampling Brownian motion, adapted to the query patterns typical to SDE solvers. The scheme produces exact samples using $\mathcal{O}(1)$ memory and average-case $\mathcal{O}(1)$ time. In particular, it can reconstruct its past trajectory, which is necessary for the use of adjoint equations. The scheme operates by combining splittable Pseudo-Random Number Generators (PRNGs), a binary tree of dependent intervals, and a Least Recently Used (LRU) cache of recent queries.

Finally, we demonstrate that the theoretical construction of adjoint SDEs (which may be used to backpropagate through an SDE) may be simplified by using the pathwise formulation of rough path theory. In particular this avoids the previous theoretical complexity of two-sided filtrations.

To facilitate the use of these techniques, we have implemented them as part of an open-source PyTorch-compatible general-purpose SDE library, [redacted] . This may be found at `https://github.com/[redacted]` .

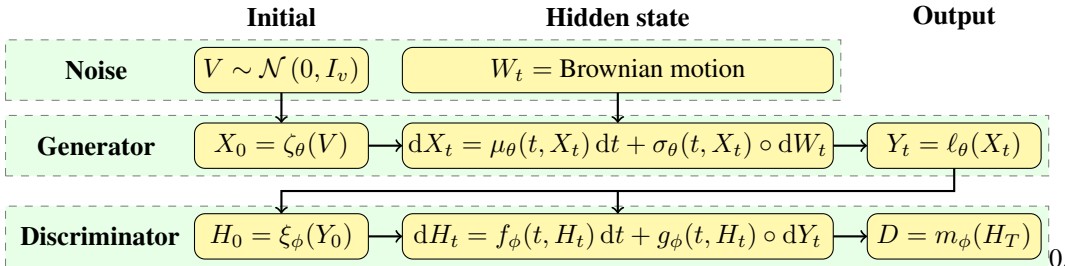

Figure 1: Summary of equations.

## 2 METHOD

### 2.1 SDEs AS GANs

Consider some "noise" distribution $\mu$ on a space $\mathcal{X}$, and a target probability distribution $\nu$ on a space $\mathcal{Y}$. A generative model for $\nu$ is a learnt function $G_\theta : \mathcal{X} \to \mathcal{Y}$ trained so that the (pushforward) distribution $G_\theta(\mu)$ approximates $\nu$. For our purposes, a Generative Adversarial Network (GAN) may then be characterised as a choice of $G_\theta$ which may be sampled from (by sampling $\omega \sim \mu$ and then evaluating $G_\theta(\omega)$), but for which the probability density of $G_\theta(\mu)$ is not computable (due to the complicated structure of $G_\theta$).

The name "adversarial" then arises from the fact that a GAN is trained by examining the statistics of samples from $G_\theta(\mu)$, most typically a learnt scalar statistic, parameterised by a discriminator. (Although variations such as MMD-GANs instead use fixed vector-valued statistics (Li et al., 2015).)

Now consider SDEs. Consider some (Stratonovich) integral equation of the form

$$X_0 \sim \mu, \quad \mathrm{d}X_t = f(t, X_t)\,\mathrm{d}t + g(t, X_t) \circ \mathrm{d}W_t,$$

for initial probability distribution $\mu$ and (Lipschitz) functions $f$, $g$ and Brownian motion $W$. The strong solution to this SDE may be described as the (unique) map $F$ such that $F(\mu, W) = X$ almost surely (Rogers & Williams, 2000, Chapter V, Definition 10.9).

Intuitively, SDEs are maps from a noise distribution (Wiener measure, the distribution of Brownian motion) to some solution distribution, which is some probability distribution on path space. We recommend any of Karatzas & Shreve (1991), Rogers & Williams (2000), or Revuz & Yor (2013) as an introduction to the theory.

SDEs can be sampled from with relative ease: this is what a numerical SDE solver does. However, evaluating its probability density is not possible; in fact it is not even defined in the usual sense.[1] This scenario – no available/tractable densities, but sampling is available – is now the familiar setting of a GAN.

Moreover, this essentially generalises the typical procedure by which a parameterised SDE is fit to data, which is usually done by matching certain statistics (such as option prices).

## 2.2 GENERATOR

Let $Z$ be a random variable on $y$-dimensional path space. Loosely speaking, this is the space of continuous functions $f \colon [0, T] \to \mathbb{R}^y$ for some fixed time horizon $T > 0$. For example, this may correspond to the (interpolated) evolution of stock prices over time. This is what we seek to model.

Let $W \colon [0, T] \to \mathbb{R}^w$ be a $w$-dimensional Brownian motion, and $V \sim \mathcal{N}(0, I_v)$ be drawn from a $v$-dimensional standard multivariate normal. The values $w, v$ are hyperparameters describing the size of the noise.

Let

$$\zeta_\theta \colon \mathbb{R}^v \to \mathbb{R}^x, \qquad \mu_\theta \colon [0, T] \times \mathbb{R}^x \to \mathbb{R}^x, \qquad \sigma_\theta \colon [0, T] \times \mathbb{R}^x \to \mathbb{R}^{x \times w}, \qquad \ell_\theta \colon \mathbb{R}^x \to \mathbb{R}^y,$$

where $\zeta_\theta$, $\mu_\theta$ and $\sigma_\theta$ are (Lipschitz) neural networks, and $\ell_\theta$ is linear. Collectively they are parameterised by $\theta$. The dimension $x$ is a hyperparameter describing the size of the hidden state.

We seek to learn a (Stratonovich) SDE of the form

$$X_0 = \zeta_\theta(V), \qquad \mathrm{d}X_t = \mu_\theta(t, X_t)\,\mathrm{d}t + \sigma_\theta(t, X_t) \circ \mathrm{d}W_t, \qquad Y_t = \ell_\theta(X_t), \qquad (1)$$

for $t \in [0, T]$, with $X \colon [0, T] \to \mathbb{R}^x$ the (strong) solution to the SDE, such that in some sense $Y \overset{\mathrm{d}}{\approx} Z$. That is to say, the model $Y$ should have approximately the same distribution as the target $Z$ (for some notion of approximate). The solution $X$ is guaranteed to exist given mild conditions (such as Lipschitz $\mu_\theta$, $\sigma_\theta$).

**Network architecture** $\quad \zeta_\theta, \mu_\theta$, and $\sigma_\theta$ may be taken to be any standard network architecture, such as a simple feedforward network. (The choice does not affect the GAN construction.)

**Hidden state** $\quad$ The solution $X$ represents hidden state, and is not the output of the model. If it were the output, then future evolution would satisfy a Markov property, of being dependent on the past only through the present, which need not be true in general.

This is the reason for the additional $\ell_\theta$ mapping to $Y$. Practically speaking, during an SDE solve, $Y$ may be concatenated alongside $X$, and $\ell_\theta$ concatenated with $\mu_\theta$.

---

[1]Technically speaking, a probability density is the Radon–Nikodym derivative of the measure with respect to the Lebesgue measure. However, the Lebesgue measure only exists for finite dimensional spaces. In infinite dimensions, it is possible to define densities with respect to for example Gaussian measures, but this is less obviously meaningful when used with maximum likelihood.

**Initial condition** It is important that there be an additional source of noise for the initial condition, passed through a nonlinear $\zeta_\theta$, as $Y_0 = \ell_\theta(\zeta_\theta(V))$ does not depend on the Brownian noise $W$.

**Stratonovich versus Itô** The choice of Stratonovich solutions over Itô solutions is not mandatory, but will turn out to be a little theoretically neater when we discuss the adjoint method in Section 3.1.

**Sampling** Given a trained model, we sample from it by sampling some initial noise $V$ and some Brownian motion $W$, and then solving equation (1) with a numerical SDE solver. Any standard numerical SDE solver may be used. As we consider Stratonovich integrals, then we use the midpoint method (which converges to the Stratonovich solution), rather than the Euler–Maruyama method (which converges to the Itô solution).

**Comparison to the Fokker–Planck equation** The distribution of an SDE, as learnt by a neural SDE, contains more information than the distribution obtained by solving a Fokker–Planck equation. The solution to a Fokker–Planck equation gives the (time evolution of the) probability density of a solution *at fixed times*. It does not encode information about the time evolution of individual sample paths. This is exemplified by stationary processes, whose distribution does not change over time.

## 2.3 DISCRIMINATOR

Each sample from the generator is a path $Y \colon [0, T] \to \mathbb{R}^y$; the discriminator must accept such paths as inputs. There is a natural choice: parameterise the discriminator as another neural SDE.

Let

$$\xi_\phi \colon \mathbb{R}^y \to \mathbb{R}^h, \quad f_\phi \colon [0, T] \times \mathbb{R}^h \to \mathbb{R}^h, \quad g_\phi \colon [0, T] \times \mathbb{R}^h \to \mathbb{R}^{h \times y}, \quad m_\phi \colon \mathbb{R}^h \to \mathbb{R},$$

where $\xi_\phi$, $f_\phi$ and $g_\phi$ are (Lipschitz) neural networks, and $m_\phi$ is linear. Collectively they are parameterised by $\phi$. The dimension $h$ is a hyperparameter describing the size of the hidden state.

Recalling that $Y$ is the generated sample, then the discriminator is an SDE of the form

$$H_0 = \xi_\phi(Y_0), \qquad \mathrm{d}H_t = f_\phi(t, H_t)\,\mathrm{d}t + g_\phi(t, H_t) \circ \mathrm{d}Y_t, \qquad D = m_\phi(H_T), \qquad (2)$$

for $t \in [0, T]$, with $H \colon [0, T] \to \mathbb{R}^h$ the (strong) solution to this SDE, which exists given mild conditions (such as Lipschitz $f_\phi$, $g_\phi$). The value $D \in \mathbb{R}$, which is a function of the terminal hidden state $H_T$, is the discriminator's score for real versus fake.

**Neural CDEs** The discriminator follows the formulation of a neural CDE (Kidger et al., 2020b) with respect to the control $Y$. Neural CDEs are the continuous-time analogue to RNNs, just as neural ODEs are the continuous-time analogue to residual networks (Chen et al., 2018). This is what motivates equation (2) as a probably sensible choice of discriminator. Moreover, it means that the discriminator enjoys theoretical properties, such as universal approximation with respect to compact sets of paths.

**Training data** Just described is how the discriminator is applied to the generator output. For the training data, the analogous thing is done, as follows.

Suppose for simplicity that we observe samples from $Z$ as an irregularly sampled but fully observed time series $\mathbf{z} = ((t_0, z_0), \dots, (t_n, z_n))$, where without loss of generality $t_0 = 0$ and $t_n = T$.

Then we may (linearly) interpolate to produce $\widehat{z} \colon [0, T] \to \mathbb{R}^y$ such that $\widehat{z}(t_i) = z_i$, and compute

$$H_0 = \xi_\phi(\widehat{z}(t_0)), \qquad \mathrm{d}H_t = f_\phi(t, H_t)\,\mathrm{d}t + g_\phi(t, H_t) \circ \mathrm{d}\widehat{z}_t, \qquad D = m_\phi(H_T)$$

as before.

Using an interpolation of the data represents an approximation to the underlying continuous-time process from which the data was observed; see Kidger et al. (2020b). Additionally the interpolation need not be linear; all that is required is to produce the distribution on path space that is desired to be modelled.

If the data is actually partially observed, has asynchronous sampling, or is of variable length, then the interpolation may still be performed in much the same way. See the examples of Kidger (2020).

**Initial condition and hidden state**  As with the generator, it is important that there be a learnt initial condition, and that the output be a function of $H_T$ and not a univariate $H_T$ itself. (See also Kidger et al. (2020b), who emphasise the need for a learnt initial condition.)

**Single SDE solve**  In practice, both generator and discriminator may be concatenated together into a single SDE solve. The state is the combined $[X, Y, H]$, the drift is the combined $[\mu_\theta, \ell_\theta \circ \mu_\theta, f_\phi \circ \ell_\theta \circ \mu_\theta]$, and the diffusion is the combined $[\sigma_\theta, \ell_\theta \circ \sigma_\theta, g_\phi \circ \ell_\theta \circ \sigma_\theta]$. Then $H_T$ is extracted from the final hidden state, and $m_\theta$ applied, to produce the discriminator's score for that sample.

**Training loss**  The training losses used are the usual one for Wasserstein GANs (Goodfellow et al., 2014; Arjovsky et al., 2017). Let $Y_\theta \colon (V, W) \mapsto Y$ represent the overall action of the generator, and $D_\phi \colon Y \mapsto D$ the overall action of the discriminator. Then the generator is optimised with respect to

$$\min_\theta \left[ \mathbb{E}_{V,W} D_\phi(Y_\theta(V, W)) \right],$$

and the discriminator is optimised with respect to

$$\max_\phi \left[ \mathbb{E}_{V,W} D_\phi(Y_\theta(V, W)) - \mathbb{E}_{\mathbf{z}} D_\phi(\widehat{z}) \right].$$

Training is performed via stochastic gradient descent techniques as usual. Backpropagation may be performed either through the internal operations of the numerical SDE solver, or via the adjoint method for SDEs (Li et al., 2020). In the latter case, then the entire SDE is treated as a single differentiable primitive within the computation graph.

**Lipschitz regularisation**  Wasserstein GANs need a Lipschitz discriminator, for which a variety of methods have been proposed. We use gradient penalty (Gulrajani et al., 2017), finding that neither weight clipping nor spectral normalisation worked (Arjovsky et al., 2017; Miyato et al., 2018).

We attribute this to the observation that neural SDEs (as with RNNs) have a recurrent structure. If a single step has Lipschitz constant $\lambda$, then the Lipschitz constant of the overall neural SDE will be $\mathcal{O}(\lambda^T)$ in the time horizon $T$. Even small positive deviations from $\lambda = 1$ produce large Lipschitz constants. Gradient penalty avoids this by regularising the Lipschitz constant of the overall network, by adding an additional regularisation term

$$\mathbb{E}_z(\|\nabla_z D_\phi(z)\|_2 - 1)^2,$$

where $z$ is sampled from all convex combinations of the true and generated distributions.

The use of gradient penalty does require a double backward. This is a concern we shall return to in Section 5.

## 2.4 Extensions

**Conditional GANs**  This approach may be extended to conditional GANs; simply append the extra context to both generator and discriminator as in Mirza & Osindero (2014); Ren et al. (2016)

**MMD-GANs**  Given a kernel on path space (not just on samples as in Briol et al. (2020)), for example the signature kernel (Király & Oberhauser, 2019; Toth & Oberhauser, 2020), then the neural SDE may alternatively be trained as an MMD-GAN as well (Li et al., 2015).

**Jumps**  It is straightforward to include jump terms (Jia & Benson, 2019) in the generator. The formulation of the discriminator is unchanged.

We consider conditional GANs in our experiments, and leave MMDs and jumps for future work.

## 3 Efficient computation

The SDE-as-GAN formulation is expected to be the primary interest for its machine learning applications.

We now provide two further technical contributions. First, we demonstrate how the construction of the adjoint equations may be performed straightforwardly via rough path theory. Second, we construct a new scheme for simulating Brownian motion, which we dub the Brownian Interval.

As a practical matter, these may be handled by SDE libraries, without the end user having to worry about how they are performed. And indeed, we make these available in the [redacted] library.

## 3.1 ROUGH ADJOINT EQUATION

Neural differential equations may be backpropagated through either by backpropagating through the internal operations of the solver, or by treating the entire neural differential equation as a differentiable primitive via the *adjoint method* (Pontryagin et al., 1962; Chen et al., 2018). The adjoint method backpropagates by solving another differential equation, the *adjoint equation*, backwards in time. See Kidger et al. (2020a) for a clear exposition in the ODE setting.

However, it does not straightforwardly extend to the SDE setting, as the theory of SDEs relies on arguments that depend on the arrow of time, such as filtrations. Li et al. (2020) manage to handle this by using a nonstandard setting involving subtle arguments with two-sided filtrations.

Here we show that the issue may instead be straightforwardly resolved by the adoption of rough path theory. Hodgkinson et al. (2020) give a very readable introduction, and Friz & Hairer (2014); Friz & Victoir (2010) are standard textbooks. In this formulation: (a) the integrals $\mathbb{W}_{s,t} = \int_s^t (W_r - W_s) \circ dW_r$ are defined probabilistically as typically Stratonovich integrals; (b) the joint probability distribution $(W, \mathbb{W})$ is sampled; (c) the solution to the SDE is defined *pathwise* with respect to the sample.[2] The solution is the same as the strong solution given by usual SDE theory, but now the probability is contained just within $(W, \mathbb{W})$, and complications such as filtrations do not appear.

It is instructive to note that this is the same procedure as performed when using numerical SDE solvers. Samples are drawn from $(W, \mathbb{W})$, and then the SDE is solved pathwise with respect to those samples. The higher order terms $\mathbb{W}$ appear for example in Milstein's method (Kloeden & Platen, 1992), originally constructed for solving Itô SDEs.

What this then means is that it is straightforward to make sense of notions like running SDEs backwards in time: we sample from $(W, \mathbb{W})$ as before, and then just traverse the sample backwards.

**Theorem (Informal).** *Consider the Stratonovich SDE of equation* (1)*, and let L be a (scalar) loss on $X_T$. Then the adjoint process $a_t = dL(X_T)/dX_t$ is a strong solution of the linear Stratonovich SDE*

$$da_t = -(a_t \cdot \nabla)\, \mu_\theta(t, X_t)\, dt - (a_t \cdot \nabla)\, \sigma_\theta(t, X_t) \circ dW_t$$

*for $t \in [0, T]$. In particular $W_t$ is the same Brownian noise as used in the forward pass.*

This is cheap to compute as it involves only vector-Jacobian products. This is equivalent to the adjoint as given by Li et al. (2020): the difference is through our use of conventional Stratonovich integrals over their "backward Stratonovich integrals" – and we argue that interpreting these as rough integrals is better still.

The proof may be found in Appendix A.

## 3.2 SIMULATING BROWNIAN MOTION

Numerically solving an SDE requires sampling Brownian motion, conditional on its previous samples. Mathematically this is straightforward. Let $W_{s,t} = W_t - W_s \in \mathbb{R}^w$. Then for $s < t < u$, Lévy's Brownian bridge (Revuz & Yor, 2013) states that

$$W_{s,t}|W_{s,u} = \mathcal{N}\left(\frac{t-s}{u-s} W_{s,u}, \frac{(u-t)(t-s)}{u-s} I_w\right). \tag{3}$$

---

[2]For almost all samples. In fact the excluded null set is the same for all SDEs. Note the use of Stratonovich integrals – this is the reason for our previous preference for them over Itô integrals.

The difficulty here is computational. On the adjoint pass, the same Brownian sample must be reconstructed, potentially at locations other than were used on the forward pass. A memory intensive approach is to store every sample, and apply equation (3) when appropriate. Gaines & Lyons (1997); Li et al. (2020) instead approach this via the "Brownian Tree". However this produces only approximations, as the real line must be discretised to some tolerance at which the tree is terminated. Practically speaking this is also slow, as small tolerances demand deep traversals of the tree; indeed Li et al. (2020) do not use it in their experiments for this reason.

We introduce the "Brownian Interval", which improves upon this with exact samples and fast query times. Similar to the dyadic tree of points used in the Brownian Tree, we now instead we have a binary tree of intervals. Each parent interval is the disjoint union of its child intervals. What is simulated is actually the increments $W_{s,t}$, not displacements $W_t$. This is because (a) this is what is actually used in an SDE solver, and (b) this is the appropriate interface when additionally desiring the higher order term $\mathbb{W}_{s,t} = \int_s^t W_{s,r} \circ \mathrm{d}W_r$, such as for Milstein's method.

The tree starts as a stump consisting of just the global interval $[0, T]$. New leaf nodes are created as queries over intervals are made. For example, making a first query at $[s, t] \subseteq [0, T]$ (an operation that returns $W_{s,t}$) produces the binary tree shown in Figure 2a; making a subsequent query at $[u, v]$ with $u < s < v < t$ produces Figure 2b. Using a splittable PRNG (Salmon et al., 2011), each child node also has a random seed deterministically produced from the seed of its parent.

The tree thus completely encodes the conditional statistics of Brownian motion: $W_{s,t}, W_{t,u}$ are completely specified by $s, t, u, W_{s,u}$, equation (3), and the random seed associated with $[s, u]$.

---

**Algorithm 1:** Sampling the Brownian Interval

**Type:** Let *Node* denote a 5-tuple consisting of an interval, a seed, and three optional *Node*s, corresponding to the parent node, and two child nodes, respectively. (Optional as the root has no parent and leaves have no children.)

**Input:** Interval $[s, t] \subseteq [0, T]$

**State:** Binary tree with elements of type *Node*, with root $\widehat{I} = ([0, T], \widehat{s}, *, \widehat{I}_{\texttt{left}}, \widehat{I}_{\texttt{right}})$. A *Node* $\widehat{J}$.

**Result:** Sample increment $W_{s,t}$

```
# The returned 'nodes' is a list of Nodes whose
# intervals partition [s, t]. Practically speaking
# this will usually have only one or two elements.
nodes = traverse(Ĵ, [s, t], [ ])

def sample(I : Node):
    if I is Î then
    |   return N(0, T) sampled with seed ŝ.
    Let I = ([a, b], s, I_parent, I_left, I_right)
    Let I_parent = ([a_p, b_p], s_p, I_pp, I_lp, I_rp)
    W_parent = sample(I_parent)
    if I_i is I_rp then
    |   W_left = bridge(a_p, b_p, a, W_parent, s)
    |   return W_parent − W_left
    else
    |   return bridge(a_p, b_p, b, W_parent, s)
sample = LRUCache(sample)

Ĵ ← nodes[−1]
return Σ_{I∈nodes} sample(I)
```

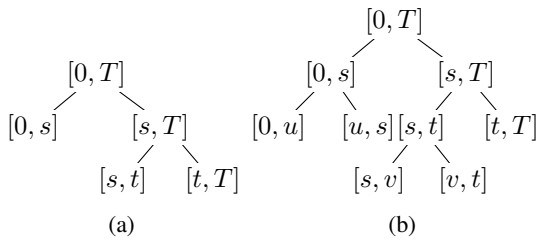

(a)                         (b)

Figure 2: Binary tree of intervals.

---

Computing $W_{s,t}$ in this way requires $W_{s,u}$, which is not itself stored; in principle it is instead calculated by recursing up the tree. This would be very slow (recursing to the root on every query), except that an LRU cache is additionally applied to the computed increments $W_{s,t}$.

Queries are exact because the tree aligns with the query points. Queries are fast because of the LRU cache; in SDE solvers, subsequent queries are likely to be close to (and thus conditional on) previous queries. The average-case (modal) time complexity is thus $\mathcal{O}(1)$. Even in the event of cache misses all the way up the tree, the worst-case time complexity will only be $\mathcal{O}(\log(1/s))$ in the average step size $s$ of the SDE solver. The (GPU) memory cost is essentially the size of the LRU cache, which is constant. There is the small additional cost of storing the tree structure itself, but this is held in CPU memory, which is for practical purposes essentially infinite.

See Algorithm 1, where `bridge` denotes equation (3), and `traverse` traverses the binary tree to find a list of nodes that are of interest, and is defined explicitly in Appendix B. Also see Appendix B for various extensions and technical considerations need to ensure this algorithm works.

## 4 EXPERIMENTS

### 4.1 DATASETS

**Stocks** We consider a dataset consisting of Alphabet/Google stock prices for 2018–2019, obtained from LOBSTER (Haase, 2013). The data consists of limit orders, in particular ask and bid prices. On average there are 605 054 values per day. Many of these do not actually change the price (specifically the midpoint or spread), so we downsample to 40 000 observations per day, specifically over the trading period 9.30am–4pm. This is then sliced into windows of length approximately one minute. We model the bivariate path consisting of the midpoint and the log-spread over this time interval.

**Weights** Next, we consider another problem that is classically produced via (stochastic) differential equations: the weight updates when training a neural network via stochastic gradient descent. We train a small convolutional network on MNIST (LeCun et al., 2010) for 100 epochs, and record its weights on every epoch. Repeated over 10 models, this produces a dataset of univariate time series; each time series corresponding to a particular scalar weight.

**Beijing Air Quality** We consider a dataset of the air quality in Beijing, from the UCI repository (Zhang et al., 2017; Dua & Graff, 2017). Each sample is a 6-dimensional time series of the $SO_2$, $NO_2$, CO, $O_3$, $PM_{2.5}$ and $PM_{10}$ concentrations, as they change over the course of a day. We train this problem as a conditional GAN, by conditioning on the one-hot encoded label for which of 14 different locations the data was measured at.

### 4.2 MODELS

We compare against the Latent ODE model of Rubanova et al. (2019) and the continuous time flow process (CTFP) of Deng et al. (2020). We used the full version of CTFPs, including latent variables.

These were selected for being competing neural differential equation models, which additionally represent differing extremes of neural SDEs. The Latent ODE model samples its noise as an initial condition, and is thereafter a pure-drift model. Meanwhile, continuous time flow processes sweep a normalising flow over a Brownian noise, and thus represent a pure-diffusion model. Neural SDEs, however, combine both drift and diffusion terms.

Between them these models cover a variety of training regimes. Latent ODEs are trained as variational autoencoders; CTFPs are trained as normalising flows; neural SDEs are trained as GANs. To our knowledge neural SDEs are the first model in their class, namely continuous-time GANs.

### 4.3 RESULTS

We study three test metrics: classification, prediction, and MMD. In each case every model is run three times and mean and standard deviation of the test metrics are reported. See Appendix C for details of hyperparameters, learning rates, optimisers and so on.

*Classification* is given by training a model to distinguish real from fake data. We use a neural CDE (Kidger et al., 2020b) for the classifier. Larger losses, meaning inability to classify, indicate better performance of the generative model.

Table 1: Classification loss. (Bold indicates best performance.)

|                      | Neural SDE              | CTFP                    | Latent ODE                    |
|----------------------|-------------------------|-------------------------|-------------------------------|
| Stocks               | **0.357 ± 0.045**       | 0.165 ± 0.087           | 0.000239 ± 0.000086           |
| Weights              | 0.507 ± 0.019           | **0.676 ± 0.014**       | 0.0112 ± 0.0025               |
| Beijing Air Quality  | 0.589 ± 0.051           | **0.764 ± 0.064**       | 0.392 ± 0.011                 |

*Prediction* is a train on synthetic, test on real (TSTR) metric (Hyland et al., 2017). We train a sequence-to-sequence model to predict the latter part of a time series given the first part. We use a neural CDE/ODE as an encoder/decoder pair. Smaller losses, meaning ability to predict, are better.

Table 2: Prediction loss. (Bold indicates best performance.)

|  | Neural SDE | CTFP | Latent ODE |
|---|---|---|---|
| Stocks | **0.144 ± 0.0446** | 0.725 ± 0.233 | 46.2 ± 12.3 |
| Weights | **0.00843 ± 0.00759** | 0.0808 ± 0.0514 | 0.127 ± 0.152 |
| Beijing Air Quality | **0.395 ± 0.056** | 0.810 ± 0.083 | 0.456 ± 0.095 |

*Maximum mean discrepancy* is a distance between probability distributions with respect to a kernel or feature map. We use the depth-5 signature transform as the feature map (Király & Oberhauser, 2019; Toth & Oberhauser, 2020). Smaller values, meaning closer distributions, are better.

Table 3: MMD loss. (Bold indicates best performance.)

|  | Neural SDE | CTFP | Latent ODE |
|---|---|---|---|
| Stocks | **1.92 ± 0.09** | 2.70 ± 0.47 | 60.4 ± 35.8 |
| Weights | **5.28 ± 1.27** | 12.0 ± 0.5 | 23.2 ± 11.8 |
| Beijing Air Quality | **0.000160 ± 0.000029** | 0.00198 ± 0.00001 | 0.000242 ± 0.000002 |

Neural SDEs produce substantially better results with respect to both the predictive (forecasting) and MMD metrics. On the stocks data, they additionally perform substantially better on the classification metric – stocks being a regime in which SDE models have classically been applied. We see that both neural SDEs and CTFPs consistently outperform Latent ODEs, which we attribute to the nature of these datasets: on these problems we expect to see some random fluctuations, and the underlying dynamics are not those of a pure drift.

## 4.4 ORNSTEIN–UHLENBECK PROCESS

We also studied an example for which the underlying distribution is known, and is given by a time-dependent Ornstein–Uhlenbeck process. See Appendix C.5.

## 5 CONSIDERATIONS

**Stochastic weight averaging**   We found that using stochastic weight averaging (Izmailov et al., 2018) was particularly helpful for improving performance, as it averages out the oscillatory training behaviour for the min-max objective used in GAN training.

**Final tanh nonlinearity**   Using a final tanh nonlinearity (on both drift and diffusion, for both generator and discriminator) constraints the rate of change of hidden state (Kidger et al., 2020b), which we found helped training.

**Lipschitz regularisation**   We found that training using the adjoint equations was difficult due to the use of gradient penalty: this involves a double backward, and thus a double adjoint, which for reasonable step sizes we found produced inaccurate enough gradients to prevent models from training. As such our experiments do not use adjoints, instead backpropagating through the solver. This is an issue that we hope may be resolved in future work.

## 6 CONCLUSION

We have shown that SDEs and GANs follow similar formalisms. Using this connection, we train *neural SDEs* as continuous time, infinite dimensional, time series GANs. Next, we introduce a new way of sampling and reconstructing Brownian motion that is both fast and memory efficient. Finally, we show that the adjoint equations may straightforwardly be developed through rough path theory.

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

# A  DERIVATION OF THE ROUGH ADJOINT EQUATION

In this section, we will present a "rough path" derivation of the adjoint equation for Neural SDEs. Since rough path theory is a well developed field, much of our analysis involves quoting key results. To begin, we recall the informal statement of the theorem that we wish to prove:

**Theorem (Informal).** *Consider the Stratonovich SDE*

$$\mathrm{d}X_t = \mu_\theta(t, X_t)\,\mathrm{d}t + \sigma_\theta(t, X_t) \circ \mathrm{d}W_t, \tag{4}$$

*where $\mu_\theta$ and $\sigma_\theta = \{\sigma_\theta^i\}_{1 \leq i \leq d}$ are sufficiently regular vector fields. Let $L$ be a scalar loss on $X_T$. Then the adjoint process $a_t = \mathrm{d}L(X_T)/\mathrm{d}X_t$ is a strong solution of the linear Stratonovich SDE*

$$\mathrm{d}a_t = -(a_t \cdot \nabla)\,\mu_\theta(t, X_t)\,\mathrm{d}t - (a_t \cdot \nabla)\,\sigma_\theta(t, X_t) \circ \mathrm{d}W_t \tag{5}$$

*for $t \in [0, T]$. In particular $W_t$ is the same Brownian noise as used in the forward pass.*

Whilst the above theorem looks simple enough, it provides us with three main challenges to address:

The first challenge in proving this theorem is that Brownian sample paths are not differentiable and thus the adjoint process will not be differentiable. In particular, we cannot use the proof given by Chen et al. (2018) where the derivative of the adjoint process is approximated using a Taylor series.

The second challenge is more subtle and relates to fact that Brownian sample paths do not have bounded variation. In particular, this means that we cannot define integrals with respect to Brownian sample paths in the Riemann-Stieltjes sense (this is discussed in Section 1.5 of Lyons et al. (2007)).

The third challenge is purely technical in that the vector fields of the adjoint equation (5) do not satisfy certain technical conditions. Typical assumptions in rough path theory are that the vector fields are either bounded (and with some smoothness) or linear. However the adjoint vector fields are linear in $a$ but nonlinear in $X$; overall they are unbounded and nonlinear. Therefore our analysis will involve separating the linear part of the adjoint equation from the bounded nonlinear part.

The outline of this section is as follows. In subsection A.1, we will derive the adjoint equation for systems where the "driving path" has bounded variation but can be non-differentiable (Challenge 1). In subsection A.2, we will discuss some aspects of rough path theory – which provides a "pathwise" integration theory for SDEs (Challenge 2). Finally, in subsection A.3, we shall put the various pieces together and derive the *rough adjoint equation* for Stratonovich SDEs (Challenge 3).

## A.1  THE ADJOINT FOR CONTROLLED DIFFERENTIAL EQUATIONS

Before we consider SDEs and Brownian motion, we first derive the adjoint equation for a slightly more manageable class of differential equation – namely the *controlled differential equation*.[3] A CDE takes a similar form to an SDE, except the system is "controlled" by a continuous path $X$ instead of Brownian motion with time (that is, we write $\mathrm{d}X_t$ instead of $\mathrm{d}t$ or $\mathrm{d}W_t$). By assuming that $X$ has bounded variation, we can use Riemann-Stieltjes integration to define well-posed CDEs (existence and uniqueness results for CDE solutions are given in Chapter 3 of Friz & Victoir (2010)).

**Theorem A.1 (Adjoint equation for CDEs that are driven by bounded variation paths).** *Consider the controlled differential equation,*

$$\mathrm{d}y_t = \sum_{i=1}^{d} f_\theta^i(y_t)\,\mathrm{d}X_t^i, \tag{6}$$

$$y_0 = \xi \in \mathbb{R}^n, \tag{7}$$

*where $X : [0, T] \to \mathbb{R}^d$ is continuous bounded variation path and each $f_\theta^i : \mathbb{R}^n \to \mathbb{R}^n$ is bounded, differentiable and with bounded first derivatives. Let $L : \mathbb{R}^n \to \mathbb{R}$ be a differentiable loss function. Then the adjoint process*

$$a_t := \frac{\mathrm{d}L(y_T)}{\mathrm{d}y_t}, \tag{8}$$

---

[3]Also referred to in the literature as a rough differential equation.

*satisfies the following linear CDE*

$$\mathrm{d}a_t = -\sum_{i=1}^{d} a_t \nabla f_\theta^i(y_t) \, \mathrm{d}X_t^i. \tag{9}$$

*Proof.* For $s \leq t$, let $\Psi_{s,t} : \mathbb{R}^n \to \mathbb{R}^n$ be the "time-reversed" flow map for the CDE (7) on $[s,t]$. So for $y \in \mathbb{R}^n$, $\Psi_{s,t}(y)$ is the solution of the CDE (7) at time $s$ so that its future value at time $t$ is $y$. Since $X$ has bounded variation, $\Psi_{s,t}$ is well-defined (via Riemann-Stieltjes integration) and satisfies

$$y = \Psi_{s,t}(y) + \sum_{i=1}^{d} \int_s^t f_\theta^i\big(\Psi_{u,t}(y)\big) \, \mathrm{d}X_u^i. \tag{10}$$

It was shown by Theorem 4.4 in Friz & Victoir (2010) that CDE flows have directional derivatives. As a result of this theorem, taking the gradient of (10) is possible and rearranging gives

$$\nabla\Psi_{s,t}(y) = \mathrm{Id} - \sum_{i=1}^{d} \int_s^t \nabla f_\theta^i\big(\Psi_{u,t}(y)\big)\nabla\Psi_{u,t}(y) \, \mathrm{d}X_u^i. \tag{11}$$

Applying the chain rule to the adjoint process $a_t = \frac{\mathrm{d}L(y_T)}{\mathrm{d}y_t}$ gives

$$a_t = \frac{\mathrm{d}L(y_T)}{\mathrm{d}y_t} = \frac{\mathrm{d}L(y_T)}{\mathrm{d}y_s}\frac{\mathrm{d}y_s}{\mathrm{d}y_t}, \tag{12}$$

where $\frac{\mathrm{d}y_s}{\mathrm{d}y_t}$ is the Jacobian matrix given by $\nabla\Psi_{s,t}(y_t)$, and so

$$a_t = a_s \nabla\Psi_{s,t}(y_t). \tag{13}$$

Thus, substituting (11) into the above yields

$$a_t = a_s - a_s\bigg(\sum_{i=1}^{d} \int_s^t \nabla f_\theta^i\big(y_u\big)\nabla\Psi_{u,t}(y_t) \, \mathrm{d}X_u^i\bigg).$$

So by the above equation along with the triangle inequality, we have

$$\bigg\|a_t - \bigg(a_s - \sum_{i=1}^{d} \int_s^t a_u \nabla f_\theta^i(y_u) \, \mathrm{d}X_u^i\bigg)\bigg\| \tag{14}$$

$$\leq \bigg\|a_t - \bigg(a_s - \sum_{i=1}^{d} \int_s^t a_s \nabla f_\theta^i(y_u) \, \mathrm{d}X_u^i\bigg)\bigg\| + \bigg\|\sum_{i=1}^{d} \int_s^t (a_u - a_s)\nabla f_\theta^i(y_u) \, \mathrm{d}X_u^i\bigg\|$$

$$= \bigg\|a_s\bigg(\sum_{i=1}^{d} \int_s^t \nabla f_\theta^i\big(y_u\big)\big(\nabla\Psi_{u,t}(y_t) - \mathrm{Id}\big) \, \mathrm{d}X_u^i\bigg)\bigg\| + \bigg\|\sum_{i=1}^{d} \int_s^t (a_u - a_s)\nabla f_\theta^i(y_u) \, \mathrm{d}X_u^i\bigg\|.$$

In order to estimate these terms, we consider the matrix-valued path $M^{t,y} : [s,t] \to \mathbb{R}^{n\times n}$ given by

$$M_u^{t,y} := -\sum_{i=1}^{d} \int_u^t \nabla f_\theta^i(\Psi_{v,t}(y)) \, \mathrm{d}X_v^i,$$

so that equation (14) becomes

$$\bigg\|a_t - \bigg(a_s - \sum_{i=1}^{d} \int_s^t a_u \nabla f_\theta^i(y_u) \, \mathrm{d}X_u^i\bigg)\bigg\| \tag{15}$$

$$\leq \bigg\|a_s \int_s^t \mathrm{d}M_u^{t,y_t}\big(\nabla\Psi_{u,t}(y_t) - \mathrm{Id}\big)\bigg\| + \bigg\|\int_s^t (a_u - a_s) \, \mathrm{d}M_u^{t,y_t}\bigg\|.$$

We use the notation $\|\gamma\|_{1\text{-var};[s,t]}$ to denote the total variation (or 1-variation) of a path $\gamma : [s,t] \to \mathbb{R}^k$,

$$\|\gamma\|_{1\text{-var};[s,t]} := \sup_{\mathcal{D}} \sum_i \|\gamma_{t_{i+1}} - \gamma_{t_i}\|,$$

where $\|\cdot\|$ is a norm on $\mathbb{R}^k$ (we use $k = d, n^2$). The supremum is taken over all partitions $\mathcal{D}$ of $[s,t]$.

It is worth noting that since $u \mapsto \nabla f_i(y_u, \theta)$ is continuous, it is bounded for $u \in [s,t]$. As a result, $M^{u,y_u}$ has bounded variation on $[s,u]$ and there exists a constant $C_1$ (depending only on $t$) such that

$$\|M^{u,y_u}\|_{1\text{-var};[s,u]} \leq C_1 \|X\|_{1\text{-var};[s,t]}, \tag{16}$$

for $u \in [s,t]$ with $s$ and $t$ sufficiently close together.

We can rewrite (11) as the following linear CDE:

$$\mathrm{d}z_u = -\big(\mathrm{d}M_u^{v,y_v}\big) z_u,$$

$$z_0 = \mathrm{Id},$$

where $z_u := \nabla \Psi_{u,v}(y_v)$ for $s \leq u \leq v \leq t$. Since the path $M^{v,y_v}$ has bounded variation, by Davie's lemma for linear CDEs (Lemma 10.56 in Friz & Victoir (2010)), there exists a constant $C_2$ such that

$$\big\|z_u - z_0\big\| \leq C_2 \|M^{v,y_v}\|_{1\text{-var};[s,v]}.$$

for $s \leq u \leq v \leq t$ whenever $s$ is sufficiently close to $t$ and we note that $z_u - z_0 = \nabla \Psi_{u,v}(y_v) - \mathrm{Id}$.

Hence by the total variation estimate (16), there exists a constant $C_3$ depending only on $t$, such that

$$\big\|\nabla \Psi_{u,v}(y_v) - \mathrm{Id}\big\| \leq C_3 \|X\|_{1\text{-var};[s,t]}, \tag{17}$$

for $s \leq u \leq v \leq t$ whenever $s$ is sufficiently close to $t$.

Since $a$ is continuous, it is bounded on $[s,t]$ and so it follows from (13) with the estimate (17) that

$$\left\| \int_s^t (a_u - a_s) \, \mathrm{d}M_u^{t,y_t} \right\| \leq \sup_{u \in [s,t]} \Big( \|a_u - a_s\| \Big) \|M^{t,y_t}\|_{1\text{-var};[s,t]}$$

$$\leq \sup_{u \in [s,t]} \Big( \|a_s \nabla \Psi_{s,u}(y_u) - a_s\| \Big) \|M^{t,y_t}\|_{1\text{-var};[s,t]}$$

$$\leq \sup_{u \in [s,t]} \Big( \|a_s\| \|\nabla \Psi_{s,u}(y_u) - \mathrm{Id}\| \Big) \|M^{t,y_t}\|_{1\text{-var};[s,t]}$$

$$\leq C_4 \|X\|_{1\text{-var};[s,t]}^2,$$

and

$$\left\| a_s \int_s^t \mathrm{d}M_u^{t,y_t} \Big( \nabla \Psi_{u,t}(y_t) - \mathrm{Id} \Big) \right\| \leq \sup_{u \in [s,t]} \Big( \|a_s\| \, \|\nabla \Psi_{u,t}(y_t) - \mathrm{Id}\| \Big) \|M^{t,y_t}\|_{1\text{-var};[s,t]}$$

$$\leq C_5 \|X\|_{1\text{-var};[s,t]}^2,$$

where the constants $C_4$ and $C_5$ only depends on $t$ (provided that $\epsilon := t - s$ is sufficiently small). Therefore equation (15) for the adjoint process becomes

$$\left\| a_t - \left( a_s - \sum_{i=1}^d \int_s^t a_u \nabla f_\theta^i(y_u) \, \mathrm{d}X_u^i \right) \right\| \leq (C_4 + C_5) \|X\|_{1\text{-var};[s,t]}^2.$$

In other words, for a fixed $t$, we have

$$a_t = a_s - \sum_{i=1}^d \int_s^t a_u \nabla f_\theta^i(y_u) \, \mathrm{d}X_u^i + O\Big( \|X\|_{1\text{-var};[s,t]}^2 \Big),$$

provided that $s$ is sufficiently close to $t$. Thus, letting $s \to t^-$ gives

$$\mathrm{d}a_t = -\sum_{i=1}^d a_t \nabla f_\theta^i(y_t) \, \mathrm{d}X_t^i,$$

as required. $\qquad\square$

## A.2 THE ROUGH PATH APPROACH TO STOCHASTIC DIFFERENTIAL EQUATIONS

In this subsection, we shall briefly outline the "pathwise solution" theory for SDEs that was made possible by the advent of rough path theory (originally proposed in Lyons (1998)). Whilst rough path theory extends beyond the SDE setting, this is not within the scope of this paper.

Let $\left(\Omega, \mathcal{F}, \mathbb{P}; \{\mathcal{F}_t\}_{t\geq 0}\right)$ be a filtered probability space containing a $d$-dimensional Brownian motion. Since the Brownian motion $W : \Omega \times [0, \infty) \to \mathbb{R}^n$ corresponds to a certain Gaussian measure on (infinite-dimensional) path space, it must be discretised in order to be used in SDE simulation. Moreover, by constructing a sequence of approximations converging to the Brownian path we can extend the adjoint equation from the bounded variation setting (see Theorem A.1) to the SDE setting.

To begin, we give a few key definitions (the signature, $p$-variation metric and geometric rough path).

**Definition A.2.** The (depth-2) *signature* of a continuous bounded variation path $X : [0, T] \to \mathbb{R}^d$ is $S^2(X) = \left\{S_{s,t}^2(X)\right\}_{0\leq s\leq t\leq T}$ where $S_{s,t}^2(X)$ is a collection of increments and integrals given by

$$S_{s,t}^2(X) := \left(1, \left\{X_t^i - X_s^i\right\}_{1\leq i\leq d}, \left\{\int_s^t \left(X_u^i - X_s^i\right) \mathrm{d}X_u^j\right\}_{1\leq i,j\leq d}\right), \tag{18}$$

where the above is defined using Riemann-Stieltjes integration.

Therefore $S^2(X) : \triangle_T \to \mathbb{R}^{1+d+d^2}$ where $\triangle_T = \{(s,t) \in [0,T]^2 : s < t\}$ is a rescaled 2-simplex.

**Definition A.3.** For $p \in [2, 3)$, the *$p$-variation metric* between functions $Z^1, Z^2 : \triangle_T \to \mathbb{R}^{1+d+d^2}$ is

$$d_p\left(Z^1, Z^2\right) := \max_{k=1,2} \sup_{\mathcal{D}} \left(\sum_{t_i \in \mathcal{D}} \left\|\pi_k\left(Z_{t_i,t_{i+1}}^1\right) - \pi_k\left(Z_{t_i,t_{i+1}}^2\right)\right\|^{\frac{p}{k}}\right)^{\frac{k}{p}}, \tag{19}$$

where $\pi_k$ denotes the projection map from $\mathbb{R}^{1+d+d^2}$ onto $\mathbb{R}^{d^k}$ (for $k = 1, 2$) and the above supremum is taken over all partitions $\mathcal{D}$ of $[0, T]$ and the norms $\|\cdot\|$ must satisfy (up to a constant)

$$\|a \otimes b\| \leq \|a\|\|b\|,$$

for $a, b \in \mathbb{R}^d$. For example, we could use the standard $L^2$ (operator) norms for vectors and matrices.

**Definition A.4.** For $p \in [2, 3)$, we say that a sequence of continuous bounded variation paths $X^N : [0, T] \to \mathbb{R}^d$ converges in the *$p$-variation* sense to a continuous map $\boldsymbol{X} : \triangle_T \to \mathbb{R}^{1+d+d^2}$ if

$$d_p\left(S^2\left(X^N\right), \boldsymbol{X}\right) \to 0, \tag{20}$$

as $N \to \infty$. When such a sequence exists, we can refer to the limit $\boldsymbol{X}$ as a geometric $p$-rough path.

We now state the following result from rough path theory (Corollary 13.22 in Friz & Victoir (2010)).

**Theorem A.5** (**Brownian motion as a geometric rough path**)**.** *Let $W$ be a standard $d$-dimensional Brownian motion and $W^N$ be the piecewise linear path with $N$ pieces that coincides with $W$ on the uniform partition $\mathcal{D}_N := \{0 = t_0 < t_1 < \cdots < t_N = T\}$ with $t_k := kh$ and mesh size $h := \frac{T}{N}$. Then there exists a random geometric $p$-rough path $\boldsymbol{W}$ $(p \in (2,3))$ such that for almost all $\omega \in \Omega$,*

$$d_p\left(S^2\left(W^N\right)(\omega), \boldsymbol{W}(\omega)\right) \to 0, \tag{21}$$

*as $N \to \infty$ for any $p \in (2,3)$ and*

$$\boldsymbol{W}(\omega) = \left\{\left(1, (W_t - W_s)(\omega), \left(\int_s^t (W_r - W_s) \otimes \circ \mathrm{d}W_r\right)(\omega)\right)\right\}_{0\leq s\leq t\leq T}. \tag{22}$$

*Hence the geometric rough path $\boldsymbol{W}$ is often referred to as* Stratonovich enhanced Brownian motion.

To put simply, this theorem states that Brownian motion can be approximated (in a rough path sense) by a sequence of bounded variation paths. This is particularly helpful within stochastic analysis as it allows one to construct pathwise solutions for SDEs governed by sufficiently regular vector fields. The central result within rough path theory that makes this possible is the *Universal Limit Theorem*. To counter the roughness of Brownian motion, this requires vector fields to have *Lip($\gamma$) regularity*.

**Definition A.6** (Lip($\gamma$) functions). A function $f : \mathbb{R}^n \to \mathbb{R}^n$ is said to be Lip($\gamma$) with $\gamma > 1$ if it is bounded with $\lfloor \gamma \rfloor$ bounded derivatives, the last being Hölder continuous with exponent $(\gamma - \lfloor \gamma \rfloor)$. Equivalently, $f$ is Lip($\gamma$) if the following norm is finite:

$$\|f\|_{\mathrm{Lip}(\gamma)} := \max_{0 \le k \le \lfloor \gamma \rfloor} \left\| D^k f \right\|_\infty \vee \left\| D^{\lfloor \gamma \rfloor} f \right\|_{(\gamma - \lfloor \gamma \rfloor)\text{-Höl}}, \tag{23}$$

where $D^k f$ is the $k$-th (Fréchet) derivative of $f$ and $\| \cdot \|_{\alpha\text{-Höl}}$ is the standard $\alpha$-Hölder norm for $\alpha \in (0,1)$. We say that $f$ is Lip(1) if it is bounded and Lipschitz continuous. That is, if the norm

$$\|f\|_{\mathrm{Lip}(1)} := \left\| f \right\|_\infty \vee \sup_{\substack{x,y \in \mathbb{R}^n \\ x \ne y}} \frac{\left\| f(x) - f(y) \right\|}{\|x - y\|}, \tag{24}$$

is finite.

**Theorem A.7** (**Universal Limit Theorem for RDEs (Theorem 5.3 in Lyons et al. (2007))**). *Let $p \in (2,3)$ and $x^N : [0,T] \to \mathbb{R}^d$ be a sequence of continuous bounded variation paths which converge in $p$-variation to a geometric $p$-rough path $\mathbf{x}$. Let $\{f_i\}_{1 \le i \le d}$ denote a collection of $\mathrm{Lip}(\gamma)$ functions on $\mathbb{R}^n$ with $\gamma > p$ and consider the controlled differential equation (CDE)*

$$\mathrm{d}y_t^N = \sum_{i=1}^d f_i(y_t^N) \, \mathrm{d}\big(x^N\big)_t^i,$$

$$y_0^N = \xi,$$

*where $\xi \in \mathbb{R}^n$ and the above differential equation is defined using Riemann-Stieltjes integration. Then there exists a unique geometric $p$-rough path $\mathbf{z} = (\mathbf{x}, \mathbf{y}) : \triangle_T \to \mathbb{R}^{1+(d+n)+(d+n)^2}$ such that $y^N$ converges to $\mathbf{y}$ in $p$-variation. Moreover, the "universal limit" $\mathbf{y}$ depends only on $\mathbf{x}$, $\{f_i\}$ and $\xi$.*

**Definition A.8.** We shall refer to $\mathbf{y}$ as the solution of the *rough differential equation* (RDE),

$$\mathrm{d}\mathbf{y}_t = \sum_{i=1}^d f_i(\mathbf{y}_t) \, \mathrm{d}\mathbf{x}_t^i. \tag{25}$$

**Remark A.9.** Theorem A.7 and Definition A.8 also apply when the vector fields $\{f_i\}$ are linear (Theorem 10.57 in Friz & Victoir (2010)).

Importantly for us, the above theory applies directly to (Stratonovich) SDEs as Brownian motion can be viewed as a geometric $p$-rough path with $p \in (2,3)$ by Theorem A.5. We refer the reader to Section 17.2 of Friz & Victoir (2010) for a detailed account of the "rough path approach" to Stratonovich theory. For our purposes, we state a *Universal Limit Theorem for Stratonovich SDEs*

**Theorem A.10** (**Remark 17.5 in Friz & Victoir (2010)**). *Suppose that $\mu$ is a $\mathrm{Lip}(1)$ function on $\mathbb{R}^{n+1}$ and $\{\sigma^k\}_{1 \le k \le d}$ are $\mathrm{Lip}(2)$ functions on $\mathbb{R}^{n+1}$. Let $\{W^N\}_{N \ge 1}$ be a sequence of piecewise linear paths converging to the Stratonovich enhanced Brownian motion $\mathbf{W}$ given by Theorem A.5. Let $\{y^N\}_{N \ge 1}$ be the sequence of solutions to the following controlled differential equations (CDEs),*

$$\mathrm{d}y_t^N = \mu(t, y_t^N) \, dt + \sum_{i=1}^d \sigma^i(t, y_t^N) \, \mathrm{d}\big(W^N\big)_t^i,$$

$$y_0 = \xi \in \mathbb{R}^n,$$

*Then $y^N$ converges in $p$-variation to a geometric $p$-rough path $\mathbf{y}$ with $p \in (2,3)$ and the process $y : [0,T] \to \mathbb{R}^n$ given by $y_t := \xi + \pi_1(\mathbf{y}_{0,t})$ coincides with the strong solution of the Stratonovich SDE*

$$\mathrm{d}y_t = \mu(t, y_t) \, dt + \sum_{i=1}^d \sigma^i(t, y_t) \circ \mathrm{d}W_t^i, \tag{26}$$

$$y_0 = \xi,$$

*almost surely.*

### A.3 THE ADJOINT FOR STOCHASTIC DIFFERENTIAL EQUATIONS

Ideally, we would like to just replace the path $X$ in Theorem A.1 with a Brownian motion (coupled with time, that is to say $(t, W_t)$.). However, that result required that $X$ have bounded variation, whilst sample paths of Brownian motion have infinite total variation. Resolving this difficulty is one of the essential reasons that rough path theory exists (see also in Section 1.5 of Lyons et al. (2007)).

As mentioned previously, the main challenge in applying rough path theory here is that the adjoint equation (29) has nonlinear unbounded vector fields, which are not $\mathrm{Lip}(\gamma)$ functions in $(a, y)$. Our trick is to first derive an adjoint equation for the stochastic process corresponding to the Jacobian (which satisfies assumptions of boundedness), and then to drive the adjoint equation by this Jacobian-valued stochastic process (which satisfies assumptions of linearity).

**Theorem A.11** (**Adjoint equation for Stratonovich SDEs**). *Suppose that $\mu_\theta$ and $\{\sigma_\theta^k\}_{1 \le k \le d}$ are bounded functions on $\mathbb{R}^{n+1}$ such that*

- *The drift vector field $\mu_\theta$ is continuously differentiable with bounded first derivative.*

- *Each noise vector field $\sigma_\theta^k$ is a $\mathrm{Lip}(\gamma)$ function with $\gamma > 2$.*

*Consider the (Stratonovich) stochastic differential equation,*

$$\mathrm{d}y_t = \mu_\theta(t, y_t)\, \mathrm{d}t + \sum_{i=1}^d \sigma_\theta^i(t, y_t) \circ \mathrm{d}W_t^i, \tag{27}$$

$$y_0 = \xi \in \mathbb{R}^n,$$

*and let $L : \mathbb{R}^n \to \mathbb{R}$ denote a differentiable loss function. Then the adjoint process*

$$a_t := \frac{\mathrm{d}L(y_T)}{\mathrm{d}y_t}, \tag{28}$$

*coincides with the strong solution of the linear Stratonovich SDE*

$$\mathrm{d}a_t = -a_t \nabla \mu_\theta(t, y_t)\, \mathrm{d}t - \sum_{i=1}^d a_t \nabla \sigma_\theta^i(t, y_t) \circ \mathrm{d}W_t^i. \tag{29}$$

*almost surely.*

*Proof.* Let $\{y^N\}_{N \ge 1}$ be the sequence of solutions to the following controlled differential equations,

$$\mathrm{d}y_t^N = \mu_\theta(t, y_t^N)\, \mathrm{d}t + \sum_{i=1}^d \sigma_\theta^i(t, y_t^N)\, \mathrm{d}(W^N)_t^i, \tag{30}$$

$$y_0^N = \xi,$$

where $\{W^N\}_{N \ge 1}$ are the piecewise linear paths converging to $W$ in $p$-variation by Theorem A.5. Hence by Theorem A.10, we have that the corresponding sequence of CDE solutions $\{y^N\}_{N \ge 1}$ converges almost surely in $p$-variation to the solution $y$ of the Stratonovich SDE (27).

Let $L$ be a differentiable loss function so that by Theorem A.1, each adjoint process

$$a_t^N = \frac{\mathrm{d}L(y_T^N)}{\mathrm{d}y_t^N}, \tag{31}$$

satisfies the linear CDE

$$\mathrm{d}a_t^N = -a_t^N \nabla \mu_\theta(t, y_t^N)\, \mathrm{d}t - \sum_{i=1}^d a_t^N \nabla \sigma_\theta^i(t, y_t^N)\, \mathrm{d}(W^N)_t^i,$$

Just as in the proof of Theorem A.1, we can rewrite the adjoint equation for $a^N$ as

$$\mathrm{d}a_t^N = -a_t^N \mathrm{d}M_t^N,$$

where the matrix-valued path $M^N : [0, T] \rightarrow \mathbb{R}^{n \times n}$ is given by

$$M_t^N := -\int_t^T \nabla \mu_\theta(s, y_s^N) \, \mathrm{d}s - \sum_{i=1}^d \int_t^T \nabla \sigma_\theta^i(s, y_s^N) \, \mathrm{d}(W^N)_s^i. \tag{32}$$

Since the vector fields $\nabla \mu_\theta$ and $\{\nabla \sigma_\theta^i\}_{1 \leq i \leq n}$ are bounded, we see that $M^N$ has bounded variation.

Recall from the universal limit theorem (Theorem A.7) that $(\boldsymbol{x}, \boldsymbol{y})$ was a geometric $p$-rough path. This carries over to our setting and thus we define the (random) geometric $p$-rough path $\boldsymbol{z} = (\boldsymbol{W}, \boldsymbol{y})$. Then by Proposition 17.1 in Friz & Victoir (2010), we have that the following rough integral exists

$$\int_0^t \varphi(W, y) \circ \mathrm{d}(W, y) = \pi_1 \left( \int_0^t \varphi(\boldsymbol{z}) \, \mathrm{d}\boldsymbol{z} \right),$$

for all $t \in [0, T]$ with probability one, provided that $\varphi = \{\varphi^i\}$ is a collection of $\mathrm{Lip}(\gamma - 1)$ functions with $\gamma > p$. Since each vector field $\sigma_\theta^i$ is $\mathrm{Lip}(\gamma)$, it follows that each gradient $\nabla \sigma_\theta^i$ is $\mathrm{Lip}(\gamma - 1)$. Therefore we can apply Proposition 17.1 in Friz & Victoir (2010) to the $\mathrm{d}W^N$ integrals in (32) and, since $\nabla \mu_\theta$ is continuous and bounded, it is clear that the $\mathrm{d}t$ integral in equation (32) also converges.

Thus, due to the regularity of $\mu_\theta$ and $\sigma_\theta$, we see that the sequence $\{M^N\}$ converges in $p$-variation to a geometric $p$-rough path $\boldsymbol{M}$ and the limiting (matrix-valued) process $M_t := \pi_1(\boldsymbol{M}_{t,T})$ satisfies

$$M_t = -\int_t^T \nabla \mu_\theta(s, y_s) \, \mathrm{d}s - \sum_{i=1}^d \int_t^T \nabla \sigma_\theta^i(s, y_s) \circ \mathrm{d}W_s^i, \tag{33}$$

almost surely. We now have all the ingredients needed to construct the rough adjoint equation (29),

1. Each CDE (30) admits a unique solution $y^N$ and the resulting sequence $\{y^N\}$ converges to the solution $y$ of the SDE (27) almost surely.

2. Each CDE (30) admits a unique adjoint process $a^N$ satisfying a linear CDE driven by $M^N$.

3. The sequence $\{M_N\}$ converges in $p$-variation to a geometric $p$-rough path (almost surely).

4. By Theorem 10.57 in Friz & Victoir (2010), we have that the Universal Limit Theorem (Theorem A.7) also holds for linear RDEs.

Therefore the sequence $\{a^N\}$ converges in $p$-variation to a geometric $p$-rough path $\boldsymbol{a}$ almost surely and the process $a_t := \frac{\mathrm{d}L(y_T)}{\mathrm{d}y_0} + \pi_1(\boldsymbol{a}_{0,t})$ coincides with the strong solution to the Stratonovich SDE

$$\mathrm{d}a_t = -a_t \circ \mathrm{d}M_t$$

$$= -a_t \nabla \mu_\theta(t, y_t) \, \mathrm{d}t - \sum_{i=1}^d a_t \nabla \sigma_\theta^i(t, y_t) \circ \mathrm{d}W_t^i.$$

almost surely. The fact that $a$ is the adjoint process follows from (31) and the continuity of $\nabla L$. $\quad\square$

**Remark A.12.** The above argument can also extend to an RDE driven by a geometric $p$-rough path. In this case, the vector fields governing the differential equation would have to be $\mathrm{Lip}(\gamma)$ with $\gamma > p$.

## B  SAMPLING BROWNIAN MOTION

### B.1  ALGORITHM

We begin with providing the complete traversal and splitting algorithm needed to find or create all intervals in the Brownian Interval, as in Section 3.2. We discuss its operation in the next section.

Here, *List* is an ordered data structure that needs to be appended to, and iterated over sequentially. For example a linked list would suffice. We let `split` denote a splittable PRNG as in Salmon et al. (2011); Claessen & Pałka (2013). We use $*$ to denote an unfilled part of the data structure, equivalent to `None` in Python or a null pointer in C/C++; in particular this is used as a placeholder

for the (nonexistent) children of leaf nodes. We use $=$ to denote the creation of a new local variable, and $\leftarrow$ to denote in-place modification of a variable.

---

**Algorithm 2:** Definition of `traverse`

---

**def** `bisect`*(I : Node, x : $\mathbb{R}$)***:**
> \# Only called on leaf nodes
> Let $I = ([a, b], s, I_{\text{parent}}, *, *)$
> $s_{\text{left}}, s_{\text{right}} = \text{split}(s)$
> $I_{\text{left}} = ([a, x], s_{\text{left}}, J, *, *)$
> $I_{\text{right}} = ([x, b], s_{\text{right}}, J, *, *)$
> $I \leftarrow ([a, b], s, I_{\text{parent}}, I_{\text{left}}, I_{\text{right}})$

**def** `traverse`*(I : Node, $[c, d]$ : Interval,* nodes *: List[Node])***:**
> Let $I = ([a, b], s, I_{\text{parent}}, I_{\text{left}}, I_{\text{right}})$
>
> \# Outside our jurisdiction - pass to our parent
> **if** $c < a$ or $d > b$ **then**
>> `traverse`$(I_{\text{parent}}, [c, d]$, nodes)
>> return
>
> \# It's $I$ that is sought. Add $I$ to the list and return.
> **if** $c = a$ and $d = b$ **then**
>> nodes.append($I$)
>> return
>
> \# Check if $I$ is a leaf or not.
> **if** $I_{left}$ is $*$ **then**
>> \# $I$ is a leaf
>> **if** $a = c$ **then**
>>> \# If the start points align then create children and add on the left child.
>>> \# (Which is created in `bisect`.)
>>> `bisect`$(I, d)$
>>> nodes.append($I_{\text{left}}$)      \# nodes is passed by reference
>>> return
>>
>> \# Otherwise create children and pass on to our right child.
>> \# (Which is created in `bisect`.)
>> `bisect`$(I, c)$
>> `traverse`$(I_{\text{right}}, [c, d]$, nodes)
>> return
>
> **else**
>> \# $I$ is not a leaf.
>> Let $I_{\text{left}} = ([a, m], s_{\text{left}}, I, I_{ll}, I_{lr})$
>> **if** $d \leq m$ **then**
>>> \# Strictly our left child's problem.
>>> `traverse`$(I_{\text{left}}, [c, d]$, nodes)
>>> return
>>
>> **if** $c \geq m$ **then**
>>> \# Strictly our right child's problem.
>>> `traverse`$(I_{\text{right}}, [c, d]$, nodes)
>>> return
>>
>> \# A problem for both of our children.
>> `traverse`$(I_{\text{left}}, [c, m]$, nodes)
>> `traverse`$(I_{\text{right}}, [m, d]$, nodes)
>> return

---

## B.2 DISCUSSION

The function `traverse` is simply a depth-first tree traversal for locating an interval within a binary tree. The search may split into multiple (potentially parallel) searches (on the last few lines) if the target interval crosses the intervals of multiple existing leaf nodes. If its target is not found then additional nodes are created if needed.

Sections 3.2 and B.1 now between them define the algorithm in technical detail.

There are some further technical considerations worth mentioning. Recall that the context we are explicitly considering is when sampling Brownian motion to solve an SDE forwards in time, then the adjoint backwards in time, and then discarding the Brownian motion. This motivates several of the choices here.

**Small intervals**  First, the access patterns of SDE solvers are quite specific. Queries will be over relatively small intervals: the step that the solver is making. This means that the list of nodes populated by `traverse` is typically small. In our experiments we observed it usually only consisting of a single element; occasionally two. In contrast if the Brownian Interval has built up a reasonable tree of previous queries, and was then queried over $[0, s]$ for $s \gg 0$, then a long (inefficient) list would be returned. It is the fact that SDE solvers do not make such queries that means this is acceptable.

**Searching from $\widehat{J}$**  Moreover, the queries are either just ahead (fixed-step solvers; accepted steps of adaptive-step solvers) or just before (rejected steps of adaptive-step solvers) previous queries. Thus in Algorithm 1, we keep track of the most recent node $\widehat{J}$, so that we begin `traverse` near to the correct location.

**LRU cache**  The fact that queries are often close to one another is also what makes the strategy of using an LRU (least recently used) cache work. Most queries will correspond to a node that have a recently-computed parent in the cache.

**Backward pass**  The queries are broadly made left-to-right (on the forward pass), and then right-to-left (on the backward pass). (Other than the occasional rejected adaptive step.)

Left to its own devices, the forward pass will thus build up a highly imbalanced binary tree. At any one time, the LRU cache will contain only nodes whose intervals are a subset of some contiguous subinterval $[s, t]$ of the query space $[0, T]$. Letting $n$ be the number of queries on the forward pass, then this means that the backward pass will consume $\mathcal{O}(n^2)$ time – each time the backward pass moves past $s$, then queries will miss the LRU cache, and a full recomputation to the root will be triggered, costing $\mathcal{O}(n)$. This will then hold only nodes whose intervals are subsets of some contiguous subinterval $[u, s]$: once we move past $u$ then this $\mathcal{O}(n)$ procedure is repeated, $\mathcal{O}(n)$ times. This is clearly undesirable.

This is precisely analogous to the classical problem of optimal recomputation for performing backpropagation, whereby a dependency graph is constructed, certain values are checkpointed, and a minimal amount of recomputation is desired; see Griewank (1992).

In principle the same solution may be applied: apply a snapshotting procedure in which specific extra nodes are held in the cache. This is a perfectly acceptable solution, but implementing it requires some additional engineering effort, carefully determining which nodes to augment the cache with.

Fortunately, we have an advantage that Griewank (1992) does not: we have some control over the dependency structure between the nodes, as we are free to prespecify any dependency structure we like. That is, we do not have to start the binary tree as just a stump. We may exploit this to produce an easier solution.

Given some estimate $\nu$ of the average step size of the SDE solver, a size of the LRU cache $L$, and *before a user makes any queries*, we simply make some queries of our own. These queries correspond to the intervals $[0, T/2], [T/2, T], [0, T/4], [T/4, T/2], \ldots$, so as to create a dyadic tree, such that the smallest intervals (the final ones in this sequence) are of size not more than $\nu L$. (In practice we use $0.8 \times \nu L$ as an additional safety factor.)

Letting $[s, t]$ be some interval at the bottom of this dyadic tree, where $t \approx s + 0.8\nu L$, then we are capable of holding every node within this interval in the LRU cache. Once we move past $s$ on the backward pass, then we may in turn hold the entire previous subinterval $[u, s]$ in the LRU cache, and in particular the values of the nodes whose intervals lie within $[u, s]$ may be computed in only logarithmic time, due to the dyadic tree structure.

This is now analogous to the Brownian Tree of Gaines & Lyons (1997); Li et al. (2020). (Up to the use of intervals rather than points.) If desired, this approach may be loosely interpreted as placing a Brownian Interval on every leaf of a shallow Brownian Tree.

**Recursion errors**    We find that for some problems, the recursive computations of `traverse` (and in principle also `sample`, but this is less of an issue due to the LRU cache) can occasionally grow very deep. In particular this occurs when crossing the midpoint of the pre-specified tree: for this particular query, the traversal must ascend the tree to the root, and then descend all the way down again. As such `traverse` should be implemented with trampolining and/or tail recursion to avoid maximum depth recursion errors.

**CPU vs GPU memory**    We describe this algorithm as requiring only constant memory. To be more precise, the algorithm requires only constant GPU memory, corresponding to the fixed size of the LRU cache. As the Brownian Interval receives queries then its internal tree tracking dependencies will grow, and CPU memory will increase. For deep learning models, GPU memory is usually the limiting (and so more relevant) factor.

**Stochastic integrals**    What we have not discussed so far is the simulation of integrals such as $\mathbb{W}_{s,t} = \int_s^t W_{s,r} \circ \mathrm{d}W_r$ and $H_{s,t} = \frac{1}{t-s}\int_s^t W_{s,r}\,\mathrm{d}r$ which are used in higher order SDEs solvers (such as the Runge-Kutta methods in Rößler (2010) and the log-ODE method in Foster et al. (2020)). Just like increments $W_{s,t}$, these integrals fit nicely into an interval-based data structure.

In general simulating the pair $(W_{s,t}, \mathbb{W}_{s,t})$ is known to be a difficult problem (Dickinson, 2007), and exact solutions are only known when $W$ is one or two dimensional (Gaines & Lyons, 1994). However, the approximation developed in Davie (2014) and further analysed using rough path theory by Flint & Lyons (2015) constitutes a simple and computable solution. Their approach is to generate

$$\widetilde{\mathbb{W}}_{s,t} := \frac{1}{2}W_{s,t} \otimes W_{s,t} + H_{s,t} \otimes W_{s,t} - W_{s,t} \otimes H_{s,t} + \lambda_{s,t},$$

where $\lambda_{s,t}$ is an anti-symmetric matrix with independent entries $\lambda_{s,t}^{i,j} \sim \mathcal{N}\left(0, \frac{1}{12}(t-s)^2\right), i < j$.

In both papers, the authors input $\widetilde{\mathbb{W}}$ into an SDE solver (the Milstein and log-ODE methods respectively) and prove that the resulting approximation achieves a 2-Wasserstein convergence rate beyond $O\left(1/\sqrt{N}\right)$, where $N$ is the number of steps. We have follow-up work planned on this topic.

## C    EXPERIMENTAL DETAILS

### C.1    GENERAL NOTES

**Code**    Our code is available at `[redacted]`.

**Software**    We used PyTorch (Paszke et al., 2019) as an autodifferentiable framework. We used the [redacted] library to solve SDEs. We used the Signatory library (Kidger & Lyons, 2020) to calculate the signatures used in the MMD metric. We used the `torchcde` library (Kidger, 2020) for its interpolation schemes, and to solve the neural CDEs used in the classification and prediction metrics. We used the `torchdiffeq` library (Chen, 2018) to solve the neural ODEs used in the classification and prediction metrics, and for the ODE components of the Latent ODE and CTFP models.

**Architectures**    By using similar differential equation models, we were able to use essentially the same parameterisation for every model's vector fields.    We used essentially the same hyperparameters for every dataset.

To recap, the neural SDE has generator initial condition $\zeta_\theta$, generator drift $\mu_\theta$, generator diffusion $\sigma_\theta$, discriminator initial condition $\xi_\phi$, discriminator drift $f_\phi$, and discriminator diffusion $g_\phi$. All of these are parameterised as neural networks.

Meanwhile Latent ODEs have an ODE-RNN encoder (with a neural network vector field) and a neural ODE decoder (with a neural network vector field). The CTFP has an ODE-RNN encoder

(with a neural network vector field) and a continuous normalising flow (Chen et al., 2018; Grathwohl et al., 2019) (with a neural network vector field) Additionally Deng et al. (2020) condition the normalising flow on the time evolution of a neural ODE of some latent state, which requires another neural network vector field.

In every case, the neural network was parameterised as a feedforward network with 2 hidden layers, width 64, and softplus activations. The drift, diffusion and vector fields, for every model, all additionally had a tanh nonlinearity as their final operation. As per Kidger et al. (2020b) we found that this improved the performance of every model.

The neural SDE's generator has hidden state of size $x$ and the discriminator has hidden state is of size $h$. These were both taken as $x = h = 96$. Note that this is larger than the width of each hidden layer within the neural networks, so that the first operation within each neural network is a map from $\mathbb{R}^{96} \to \mathbb{R}^{64}$. Somewhat anecdotally, we found that taking the state to be larger than the hidden width was beneficial for model performance.[4]

The Latent ODE likewise has evolving hidden state, which was also taken to be of size 96.

The Latent ODE samples noise from a normally distributed initial condition, we took to have 40 dimensions. The CTFP samples noise from a Brownian motion, which as a continuous normalising flow has dimension equal to the number of dimensions of target distribution.

The neural SDE samples noise from both a normally distributed initial condition and a Brownian motion. We took the initial condition to have 40 dimensions. The number of dimensions of the Brownian motion was dataset dependent, see below.

The CTFP included a latent context vector as described in Deng et al. (2020). This was taken to have 40 dimensions.

These hyperparameters were selected based on informal initial experiments with all models.

**SDE solvers**    The SDEs used the midpoint method, without adaptive stepping. Recall that the target time series data was regularly sampled and linearly interpolated to make a path. We took the SDE solver to take a single step between each output data point.

**ODE solvers**    The ODEs of the Latent ODE and CTFP models were solved using the midpoint method, for consistency with the SDE solvers.

**CDE solvers**    The CDEs of the classification and prediction models were solved by reducing to ODEs as in Kidger et al. (2020b) and then using the midpoint method, for consistency with the SDE solvers.

**Optimisers**    The CTFP, Latent ODE, and the generator of the neural SDE were all trained with Adam (Kingma & Ba, 2015) with a learning rate of $4 \times 10^{-5}$. The discriminator of the neural SDE was trained with RMSprop with a learning rate of $4 \times 10^{-5}$. The learning rates were chosen by starting at $4 \times 10^{-4}$ (arbitrarily) and reducing until good performance was achieved. (In particular seeking to avoid oscillatory behaviour in training of the neural SDE.)

**Training**    Every model was trained for 100 epochs. The discriminator of the neural SDE received five training steps for every step with which the generator was trained, as is usual; the number of epochs given at 100 is for the generator, for a fair comparison to the other models.

Batch sizes were picked based on what was the largest possible batch size that GPU memory allowed for; these vary by problem and are given below.

**Normalisation**    All data was normalised to have zero mean and unit variance.

---

[4]This has some loose theoretical justification: a signature is a linear differential equation with very large state, and it is a universal approximator. (See Kidger et al. (2020b, Appendix B) and references within – this is a classical fact within rough analysis.) That is to say, it is a simple vector field with a large state, rather than a complicated vector field with a small state.

**Classifier and predictor** The classifier was taken to be a neural CDE with hidden state of size 32, and whose vector field was parameterised as a feedforward neural network with 2 hidden layers of width 32, with softplus activations and final tanh activation.

The predictor was taken to be a neural CDE/neural ODE encoder/decoder pair. Both had a hidden state of size 32, and vector fields parameterised as feedforward neural network with 2 hidden layers of width 32, with softplus activations and final tanh activation. 32 dimensions were used at the encoder/decoder interface.

The learning rate used was $10^{-4}$ for both models, for every dataset and generative model considered, with the one exception of CTFP on Beijing Air Quality, where we observed divergent training of the classifier; the learning rate was reduced to $10^{-5}$ for this case only.

In all cases they were trained for 50 epochs using Adam, with early stopping if the model failed to improve its training loss over 20 epochs.

The classifier took an 80%/20% train/test split of the dataset given by combining the underlying dataset and model-generated samples of equal size.

## C.2 STOCKS

Each sample is of length 100.

The batch size was 2048 for every model.

For the neural SDE, the discriminator received 1 epoch of training before the main training (of both generator and discriminator simultaneously) commenced. The weight averaging (over both generator and discriminator) was over every training epoch. The Brownian motion from which noise was sampled had 3 dimensions.

The prediction metric was based on using the first 80% of the input to predict the last 20%.

## C.3 WEIGHTS

Each sample is of length 100. Each sample corresponds to the trajectory of a single scalar weight, epoch-by-epoch, as a small convolutional model is trained on MNIST for 100 epochs. Every weight from the network is used, and treated as a separate sample. This is repeated 10 times. If $P$ is the number of parameters in the convolutional network, then the overall size of the dataset is now $(samples = 10P, length = 100, channels = 1)$.

The batch size was 4096 for the neural SDE and latent ODE. This was reduced to 1024 for the CTFP, which we found to be a very memory intensive model on this problem.

For the neural SDE, the discriminator received 10 epochs of training before the main training (of both generator and discriminator simultaneously) commenced. The weight averaging (over both generator and discriminator) was over every training epoch. The Brownian motion from which noise was sampled had 3 dimensions.

The prediction metric was based on using the first 80% of the input to predict the last 20%.

## C.4 BEIJING AIR QUALITY

Each sample is of length 24.

The data was normalised to have zero mean and unit variance.

The batch size was 1024 for every model.

For the neural SDE, the discriminator received 10 epochs of training before the main training (of both generator and discriminator simultaneously) commenced. The weight averaging (over both generator and discriminator) was over the final 40 epochs of training. (We realised that this was an obvious improvement over averaging every epoch, as was done for the previous two experiments.) The Brownian motion from which noise was sampled had 10 dimensions.

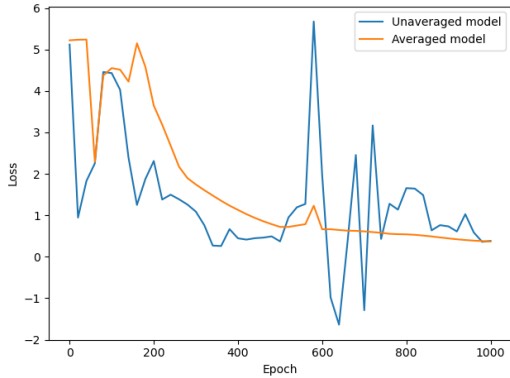

Figure 3: Loss over training for the time-dependent Ornstein–Uhlenbeck example.

The prediction metric was based on using the first 50% of the input to predict the last 50%. (An accidental change from the 80%/20% split used in the other experiments; this was kept as it is fair, as it is the same for all models on this dataset.)

## C.5 ORNSTEIN–UHLENBECK PROCESS

**Target SDE**   As an additional example for which the true underlying distribution is known, we consider the problem of training a neural SDE to match this known SDE.

The target SDE is a time-dependent Ornstein–Uhlenbeck process, of the form

$$\mathrm{d}z_t = (\mu t - \theta z_t)\mathrm{d}t + \sigma \mathrm{d}W_t.$$

We specifically take $\mu = 0.02, \theta = 0.1, \sigma = 0.4$, and generate 8192 samples from $t = 0$ to $t = 63$, sampled at every integer.

Unless otherwise specified, this example followed the same procedure as for the other experiments; for example using small feedforward neural networks to parameterise the required neural networks.

**Architectures**   The neural networks had a single hidden layer of width 16, the size of the hidden state (for both generator and discriminator) was taken to be 32, the Brownian motion was 3-dimensional, and the initial noise was 5-dimensional.

**Optimisation**   The model was trained for 1000 epochs. As with the main text, we found stochastic weight averaging to be important to obtain good performance. This was performed by averaging (without weighting) over all epochs from epoch 60, so as to use a small warm-up period.

**Loss curves**   Consider the loss

$$\min_\theta [\mathbb{E}_{V,W} D_\phi(Y_\theta(V, W)) - \mathbb{E}_\mathbf{z} D_\phi(\widehat{z})],$$

which is the generator's loss with the normalisation term on the true distribution included, and corresponds to the Wasserstein distance between distributions. The notation used is as used in Section 2.3 of the main text.

We measure this every 20 epochs and plot how it varies. See Figure 3. We see that the loss is very unstable for the unaveraged model. This highlights the importance of stochastic weight averaging, as described in Section 5.

**Sample paths**   We plot 10 samples from the true distribution against 10 samples from the learnt distribution (and examined several such plots to be sure that these were representative samples). See Figure 4.

We see that the model has indeed learnt to reproduce the true distribution. It does particularly well near the middle; it still has room to improve near $t = 0$ and $t = 63$.

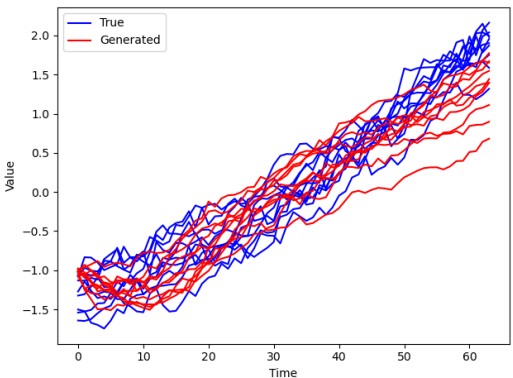

Figure 4: Sample paths from the time-dependent Ornstein–Uhlenbeck SDE, and from the neural SDE trained to match it.

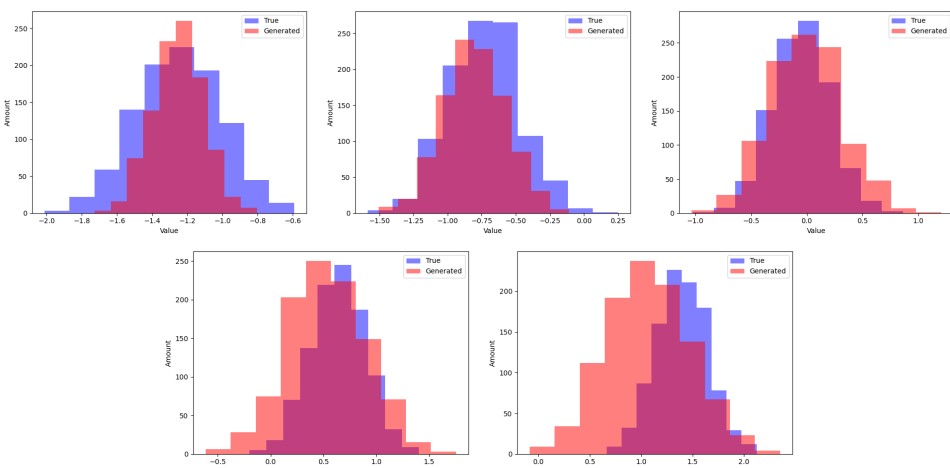

Figure 5: Top to bottom, left to right: marginal distributions at $t = 6, 19, 32, 44, 57$.

**Marginal distributions** Next we plot its marginal distributions at $t = 6, 19, 32, 44, 57$. (Corresponding to approximately 10%, 30%, 50%, 70% and 90% of the way along.) See Figure 5.

We see that the generated marginal distributions generally match the true marginal distributions. These observations match the samples previously generated; in both cases the generated distribution is a little too low at later times.

**Summary** This relatively small synthetic example demonstrates that neural SDEs can recover distributions known to be generated by SDEs. Further improvement could likely be obtained by using larger neural networks, and by training for longer.

