# OpenReview forum: "Neural SDEs Made Easy: SDEs are Infinite-Dimensional GANs"
_ICLR.cc/2021/Conference — Reject_

### Official Review · AnonReviewer1 · 2020-10-14
**An interesting idea, but with poor presentation, especially not focus on the `SDE as GAN view.**

**Rating:** 4
**Confidence:** 4

**Review:**

This paper connects SDEs and GANs and proposed to learn the drift and diffusions in SDE under the framework of GAN. The authors also show how to efficiently simulate the adjoint process and sample the Wiener process.

Though the overall idea is interesting, I have several concerns:

1. I don’t find the library [redacted] that mentioned several times in the paper.
2. Will it be problematic if the real data is not uniformly sampling across [T] and maybe sparse in some interval {t_1, t_2}? How should we do the interpolation of z in the training data paragraph and why should we linearly interpolate? I feel the description in this part is vague. Can the authors give more formal description and give some intuitions on why should we do like that theoretically?
3. I feel the Section 3 is more related to computational issue rather than the GAN formulation of neural SDE and can be individual interest? I would like to know more on the properties of the proposed GAN formulation of neural SDE, and I suggest that the authors summarizing the efficient computation part into another single paper, and focus more on the neural SDE as GAN in this paper.
4. I would like to see more show-cases on the performance of the proposed algorithm on learning given SDE with some simple drift and diffusion term if possible, that may better demonstrate the effectiveness compared with the dataset with unknown drift and diffusion. Also, some generative results is preferred, if there’re proper settings (e.g. some video scenarios), rather than the prediction results in the table.

From the current presentation, I does not find sufficient motivation on `why viewing SDE as GAN is good`. I would like to see the detailed motivation, detailed description of methods, at least high level insights that this formulation is beneficial, and sufficient experiments that show the effectiveness of the proposed method. And as a result, I think the current version is not ready for publication.

---

> ### Author Response · Authors · 2020-11-13
> **All concerns addressed.**
>
> Thank you for your review.
>
> **Regarding your concerns:**
>
> 1. It is not completely clear to us what your concern about the library is? The library is deliberately redacted for anonymity.
>
> 2. We agree that the interpolation section was not sufficiently clear, and as such have expanded this section. (We follow Kidger et al. (NeurIPS 2020), who consider exactly this problem, but missed off the reference by mistake.)
>
> 3. The discussion in Section 3 is critical for implementing these models in practice (and not just in theory). For example the introduced Brownian Interval is orders of magnitude faster than the Brownian Tree previously introduced in Li et al. (AISTATS 2020).
>
> 4. We considered adding such a synthetic example, but in our experience most reviewers claim to find them unconvincing -- as it is not surprising that for example a neural SDE could be used to model another SDE. Meanwhile our results reflect target applications for which SDEs are known to be of interest, such as financial stocks. Video generation is a wholly different topic. (e.g. the dynamics are not then continuous in pixel space.)
>
> **Regarding the motivation:**
> We have greatly expanded Section 2.1, adding additional commentary to explain this.
>
> In brief -- this is a direct extension of currently established practice.
>
> SDEs have typically been fitted to data by matching certain desired statistics, such as the observed market price of an option, in the finance industry. The discriminator of a GAN is then simply a learnt statistic -- learnt to be as difficult as possible to match. (So that at least in principle all statistics now match.) This is now the same distinction made in the mainstream ML literature between MMD-GANs and (Wasserstein) GANs. We now cover this in a lot more detail in Section 2.1.
>
> **Summary:**
> We hope that this addresses the reviewer's concerns for an increased score. We would be very happy to address any further concerns or queries they may have.

---

> > ### Comment · AnonReviewer1 · 2020-11-15
> > **Thanks for your response. Below are my ideas.**
> >
> > 1. At the first glance I think "redact" is the name of the library. Generally we will give a anonymous link in the paper rather than redacted in this way. I'm finally aware that the codes are in the supplementary materials. Sorry for that.
> > 2. Thanks for including the references on the interpolation. I would like to ask, given an SDE and we simulate the SDE and give the observation at $t_1, \cdots, t_n$ as the training data, then when we do linear interpolation, will it be far from the simulated path or will it be likely to emerge in the diffusion process. Moreover, will it influence the training of the model? Can the authors give some interpretation on that?
> > 3. I agree with the authors that Section 3 is crucial for implementation in practice, but what I would like to say is, it does not really help me to understand "why viewing SDEs as GANs is good". I would like to see more discussion on the motivation of the GANs view of SDE, as I feel this is the main argument of this paper.
> > 4. I understand on the synthetic experiment. It's not my main concern but it can improve the quality of paper in my opinion. At least it shows we can use some data-driven methods to simulate the SDE effectively with good quality.
> >
> > I appreciate the revision in Section 2.1, but I still think there can be some improvement on the paper presentation. The current paper gives me a feeling like, we tell you that with such architecture, such training loss, such training technique, we will have a good result. I feel the paper can be organized in a better way. For example, the authors can give some intuition on which property of SDEs gives the motivation of the design of the generator and discriminator, which can make the paper more readable.

---

> > > ### Author Response · Authors · 2020-11-16
> > > **Response**
> > >
> > > Thank you for your response.
> > > 1. (resolved)
> > > 2. Yes, piecewise linear interpolation of solutions stay close to solutions, so this should not be an issue. This follows immediately from the Holder continuity of solutions to SDEs.
> > > 3. The rationale for SDEs-as-GANs is that: (a) SDEs and GANs share similar properties (possibility of sampling; lack of density), and (b) the training procedure for SDEs is already very close to that of GANs - matching known statistics rather than learn statistics. For these two reasons SDE-as-GANs is a clear extension of current practice.
> > > 4. We believe that experiments on real problems with actual datasets are more convincing of the fact that our method is worthwhile.
> > >
> > > We think the presentation already provides the requested intuition. The generator is an SDE -- because that is the whole interest in neural SDEs in the first place -- whilst we provide intuition for the discriminator in terms of neural CDEs (Kidger et al, NeurIPS 2020). Similarly, we emphasise the importance of hidden state (because of the Markov property), the initial noise (so that general initial distributions may be learnt), the use of gradient penalty (as the only effective way of enforcing the Lipschitz property), and so on. At every stage we are careful to try to provide intuition.
> > >
> > > What do you think?

---

> > > > ### Comment · AnonReviewer1 · 2020-11-16
> > > > **Thanks for your quick response. Here's my idea.**
> > > >
> > > > Regarding the interpolation: I partly agree that if the interval length is small, then that's not an issue. But I'm not fully convinced that when there can be some large interval (i.e. $t_{i} - t_{i-1}$ is large), then that's still reasonable.
> > > > Regarding the experiments: The main motivation for the experiments on simulating the given SDE is that, we know the underlying SDE and thus we can better see the simulation results, for example, a maybe not so valid question is given a Langevin diffusion, can the proposed methods converge to the target distribution. For me that is more convincing.
> > > > Regarding the presentation: I would clarify that my idea for the presentation is not saying that the including contents are not necessary -- indeed they are necessary as the building blocks for the paper. But I just feel that they are only irrelavant techniques that used in the paper. From my point of view I would rather argue that one paper should focus on some main technical contribution, and leave the other parts as the implementation details. For example, for your paper I may organize as follows:
> > > >
> > > > 1. Why should we use this perspective.
> > > > 2. Under which condition the proposed methods can recover the underlying SDE (which should the main point we want to use the proposed methods).
> > > > 3. The simulation results on some known underlying SDE.
> > > > 4. The simulation results on real datasets.
> > > >
> > > > I'm happy to have more discussion on that.

---

> > > > > ### Author Response · Authors · 2020-11-20
> > > > > **All feedback applied**
> > > > >
> > > > > Thank you for being happy to have a discussion on this. Apologies for the delay in responding; we were running additional experiments (see below).
> > > > >
> > > > > **Regarding the interpolation**
> > > > > We agree that problematic data might cause issues. As a result of this conversation, the paper now specifies in "Training data" (page 4) that the data and interpolation together induce a distribution on path space, and that the interpolation should be chosen such that this is the distribution desired to be modelled.
> > > > >
> > > > > **Regarding the experiments**
> > > > > Rather than debating this back and forth, we have now included results on a synthetic task, given by a time-dependent Ornstein-Uhlenbeck process. (This is the reason for the delay in our response.) We train a neural SDE to match this example. We then plot learnt sample paths, investigate the evolution of its margnial distributions, and look at its loss curves during training. (For example highlighting the importance of using stochastic weight averaging.)
> > > > >
> > > > > **Recovering the underlying SDE**
> > > > > The loss used in a (Wasserstein) GAN is the Wasserstein metric; as a metric it has a unique global minimiser corresponding to the target distribution. For this reason it is immediate that the proposed method can recover the underlying SDE in the infinite data limit.
> > > > >
> > > > > **Regarding the presentation**
> > > > > The details such as hidden state, gradient penalty, and so on, are definitely not "irrelevant". These are explicitly stated because getting these right is part of the main technical contribution. For example the use of hidden state is a modelling choice, not an implementation detail.
> > > > >
> > > > > We (the authors) actually had a conversation about whether to split out the Brownian Interval and adjoint method into a separate paper. We unanimously decided we would rather write a single strong paper than two weak papers, as all of the material relates to each other.
> > > > >
> > > > > **Summary**
> > > > > Thank you again for your feedback. We hope we have addressed every concern that has been raised, and have updated the paper appropriately.

---

### Official Review · AnonReviewer4 · 2020-10-27
**SDEs are not GANs, but this is an interesting paper.**

**Rating:** 5
**Confidence:** 4

**Review:**

#### Summary

The authors provide a view into neural SDE models, where they mix standard (classical) SDE theory with contemporary neural SDE methods. Overall the paper aims to introduce a generic and user-friendly approach to neural SDEs, with three distinctive contributions (i) interpreting that the mathematical formulation of SDEs is directly comparable to the ML formulation of GANs, (ii) introducing a new way of sampling Brownian motion realisations, and (iii) simplifying the construction of adjoint SDEs by a pathwise formulation.

The paper is theoretically sound, avoids typical pitfalls in presenting SDE theory in ML papers, and is easy to follow. The authors cite background work in good detail and show that they are aware of the relevant related work and theory in SDE models. The paper is topical, and the theme should be of interest for the audience of ICLR.

My initial score reflects the concern points that are listed in more detail below, where, however, some of the concerns should be easy to address in updated versions of the paper.


#### Concerns

1. Novelty. Even if I found the paper interesting, I can not quite agree with the novelty statement. I found the statement (i) just simply misleading (see #2) and I recommend that you would consider revising this. The construction of (ii) is interesting, but the idea itself seems very closely related to that of Li et al. (2020), which makes it less exciting. The technical details related to (iii) are interesting. The overall novelty is limited, even if the paper is well presented.

2. SDEs are not GANs. The basic idea in Generative Adversarial Networks (GANs) lends itself directly to other generative adversarial models by swapping the generative model and the discriminator. A stochastic differential equation could typically be seen as a generative model, given a drift and diffusion. In the current form, framing the inference as a GAN feels overly complicated and does not help facilitate understanding. (Minor: There is also a typo in the paper title related to this point, 'SDEs' vs. 'GANS').

3. Practical impact not reflected in experiments. The experimental validation is based on three rather simple data sets. No comparison of solvers, nor detailed comparisons to the other methods are included.

4. Fokker-Planck or forward Kolmogorov equation formulation. I expected the Fokker-Planck equation to be discussed as a solution concept or at least mentioned in Sec. 2.

5. Use of the Stratonovich form. Throughout the paper you present Stratonovich SDEs rather than Itô SDEs which could be regarded more standard in most related ML publications. The benefits of the Stratonich form are only briefly covered, and adding details (from an Itô perspective) to Sec. 3.1 and 3.2 would do the paper good.

6. Presentation. The paper is much like a first draft. This shows in the many single-sentence paragraphs, list-like presentation, and additional details needed (in addition to brief experiments, see #3).

---

> ### Author Response · Authors · 2020-11-13
> **SDEs are GANs! Can we convince you?**
>
> Thank you for your review.
>
> Regarding your concerns:
>
> 1./2. Is the reviewer's concern that an SDE could be interpreted as any kind of generative model, not just a GAN? Both GANs and SDEs satisfy (a) efficient sampling, (b) no available/tractable density, (c) are a map from a noise distribution to a target distribution.
>
> The same cannot be said of VAEs or normalising flows, for example -- both are typically optimised by optimising a (variational bound on) the log-likelihood. This requires densities, which are not available in this context. This is what motivates this approach.
>
> We have greatly expanded Section 2.1 to help make this clear. As this seems like a sticking point we would be very happy to discuss this further.
>
> Regarding the Brownian sampling: the Brownian Tree introduced by Li et al. (2020) is unfortunately very slow -- whilst our introduced Brownian Interval is orders of magnitude faster. In our experiments we found that this made the difference between practicality and impracticality. We have added a comment to reflect this.
>
> 3. Further comparisons could be introduced; but this is always true. The datasets used are not simple -- the modelling of stocks, and understanding the training of neural network weights, are both active areas of current research.
> 4. We agree that discussion of the Fokker--Planck equation could be included, and have now done so.
> 5. We also agree that further discussion on the use of the Stratonovich form could be included, and once again have now done so.
> 6. This is perhaps a difference of opinion, but the presentation style was very deliberately chosen. Within academic writing we refer to for example "Neural Ordinary Differential Equations" (Best paper NeurIPS 2018) as a well-regarded example of the short sentence, list-like presentation. Outside of academia, this is also the same format used by the BBC (British Broadcasting Corporation).
>
> **Summary:**
> We hope that this discussion and these changes address the reviewer's concerns for an improved score, in particular with regards to the SDEs-as-GAN formulation. We are of course very happy to discuss this further.

---

### Official Review · AnonReviewer2 · 2020-10-29
**interesting connection betwee SDEs and GANs**

**Rating:** 6
**Confidence:** 4

**Review:**

# General statements

This paper introduces an interesting parallel between SDEs and GANs, and pushes the analogy to its practical implications as a way to learn neural SDEs.

Globally, I found the paper a very good read, although it sometimes lack the details that could be useful for a non-specialist. This could and should easily be corrected by a few sentences here and there.

Although I see that everything is included in the supplementary material to actually reproduce all experiments. Still, I believe that there could be some improvements to do. In particular:
  - I think that the main text / the supplementary could be augmented with a short mention regarding the network structures. even when reading the code, it is not clear how time is handled (since the nets input not only tensors like X_t or H_t, but also time). Should I understand that the raw time stamp is simply concatenated to the other input ?
  - you didn't clearly mention all the tricks and experiments you tried out. It is not clear to me to what extent the performance you report depends on the network structures you picked.


All in all, I recommend acceptance.

----
EDIT:
after seeing all reviews, and most importantly thinking about it and pondering the answers given by the authors, I am sorry that I must lower my score.
I still like the paper, that could be accepted in my opinion, but I think it oversells some contributions that are unfortunately not exploited (the brownian interval thing in particular, or am I wrong ?)
----

## Introduction
Her are some comments along the way:
* I could regret that no general background is given for the curious reader that is not already a specialist in SDE or even ODE

## SDEs as GANs
* please explain the "Initial condition" statement better: why is it important that there be an additional source of noise here ?
* You are mentioning X and H as the (strong) solutions to your SDEs (1) and (2). Are they guaranteed to exist ? I guess the Lipschitz condition you assumed is enough for this. Is that the case ?
* In the "training data" item, H_0 is a function of Y_0. i/ is this Y_0 defined as above in "initial condition" ? ii/ Is there a reason H_0 is not a function of z_0 ? iii/ It makes the decision D done on training data actually to depend on \theta, and not only on \phi (through Y_0=l_\theta(\zeta_\theta(V)). is that ok ? The item "initial condition and hidden state" does not make that point clearer to me.
* Could you briefly describe gradient penalty, instead of only refering to (Gulrajani 2017) ? That would make the paper more self-contained

## Efficient computation
* Section 3.1 (rough adjoint equation) is harder for me. I'm ok with the adjoint equation. Then, forgive me but I'm more uncomfortable with the (W, \mathbb{W}) couple. What is meant exactly by "sampling" them ? It means drawing (s,t) and computing the related (W, \mathbb{W}) ? For each, you compute the solution to the SDE ?
* now, assuming you get your a_t process. How do you actually use it to perform optimization ? Are you computing the gradient of the parameters wrt a_t and then averaging over time ? Basically, I need some more information on the general scheme to understand 3.1, assuming the adjoint equation is understood.

## Experiments

* The "weights" dataset is not super clear. Is the data actually a: 10xPx100 tensor, where P is the number of parameters ? (what is the value of P ?) Just to make sure: the same net is trained for all weights (what I assume), or is it a different net per weight ?

## Considerations
* in the "lipschitz regularisation" of your 2.3 section, you mention using gradient penalty, requiring adjoint, etc. But here, I understand that you actually didn't use these sophistications that were introduced in section 3.1 ? I think you should rephrase a bit here and there to actually better reflect these findings.

## References
* References are inconsistent. Sometimes abreviations, sometimes full names.
* Françios-Xavier -> François-Xavier

---

> ### Author Response · Authors · 2020-11-13
> **Suggestions implemented!**
>
> Thank you for your thorough review.
>
> We are very pleased to hear that the paper is regarded positively. Our response is broken down according to the same sections.
>
> **General statements**
> - In our experiments the network structures were parameterised as MLPs, for which all inputs were indeed concatenated. This is actually independent of the rest of the constructions (any network architecture would do), but we have added a remark to help with the reader's intuition on this point.
>
> - Tricks and experiments: we're not certain which parts the reviewer does not feel are clearly mentioned; if these can be highlighted then we would be happy to address these. Regarding the network structures in particular, however: these were MLPs, as is standard across the neural differential equation literature -- sensible choices of more complicated network structures remains an open question not just for neural SDEs, but for most other neural differential equation applications as well.
>
> **Introduction:**
> - We have updated the paper with additional references to material on SDEs. Regardless, we believe that familiarity with SDEs is going to be a prerequisite for using neural SDEs.
>
> **SDEs as GANs:**
> - The initial output $Y_0$ is independent of the Brownian noise. If there was no initial noise then $Y_0$ would be fixed, and could not in general match the data distribution.
>
> - Yes, the solutions X and H are guaranteed to exist subject to mild (Lipschitz) conditions on the vector fields. We have added a note on this.
>
> - The use of $Y_0$ in "Training data" is a typo - thankyou. This has been corrected to $\widehat{z}(t_0)$ instead.
>
> - We have now included a description of gradient penalty.
>
> **Efficient computation:**
> - To be explicit: The Brownian motion $(s, t, \omega) \mapsto W_{s, t}(\omega)$ is a collection of random variables indexed by $s, t$, representing a time a time increment; the randomness is the $\omega$. A sample then corresponds to a particular choice of $\omega$. (The same statement is true of $\mathbb{W}$.) This is the notion of sampling usually used with Brownian motion.
>
> - $a_t$ in fact _is_ the gradient on the parameters. The use of adjoint equations corresponds to continuous-time backpropagation, typical in neural differential equations. See for example "Neural Ordinary Differential Equations" (NeurIPS 2018) for the simpler ODE equivalent. Solving this SDE from the end time to the start time is directly analogous to performing backpropagation from the end of a neural network to the start.
>
> **Experiments:**
> - The data is a tensor of shape (samples=10P, length=100, channels=1); the same network architecture is trained each time. We have updated the discussion in the appendix to reflect this.
>
> **Considerations:**
> - The use of adjoints with gradient penalty is indeed tricky -- this is a limitation that we hope may be improved in future work. The adjoint derivation is included as a topic of pre-existing theoretical interest.
>
> **References:**
> - Thankyou, we have updated these.
>
> We would be very happy to address any further questions or comments the reviewer may have.

---

> > ### Comment · AnonReviewer2 · 2020-11-21
> > **Actually using backprop through solver makes the story inconsistent**
> >
> > I won't comment on the considerations raised by the other reviewers, because you have ongoing conversations with them on these.
> >
> > My strongest concern, that I feel has not been addressed (can it be ?) is about the fact that you're actually backpropagating through the solver in the end, regardless of all the theory you develop for adjoint, which actually form a large part of your paper.
> >
> > This is a pity, because I really like this brownian interval thing. But all in all, isn't the paper clearly overselling considerations that don't make it to the experiments ? Something is thus wrong somewhere in the way you tell the story. We are left trying to understand contributions/explanations that are not used / don't work : it's a waste of time for the readers...
> >
> > I don't know how the other reviewers actually feel about this, they don't seem to complain too much. Maybe this is all a detail ? I wouldn't agree : this part of training (how you backprop) is important. Your modifications clearly are not enough to remove this weak point.
> > Maybe you just push all this discussion about adjoint to some appendix ? That would focus the point much more on your main claim, which is the analogy between SDEs and GANs ? I feel a bit confused about what you should do about this.

---

> > > ### Author Response · Authors · 2020-11-21
> > > **Thanks for your response.**
> > >
> > > We completely appreciate the concern; it would have been ideal to be able to apply adjoints in the experiments.
> > >
> > > We decided to go with this presentation because we felt that all of the content of the paper was still of sufficient interest. Moreover, the theory still carries through: for example this could be resolved just by using much smaller step sizes in the solver.
> > >
> > > In brief: the main part of the paper presents SDE-GANs, adjoints, and the Brownian Interval. That our experiments only concern the first one does not diminish the contributions of the other two. Experiments need not be the denouement of a paper.

---

### Official Review · AnonReviewer3 · 2020-10-30
**Review of Neural SDEs made easy**

**Rating:** 3
**Confidence:** 3

**Review:**

Summary: this paper claims to show that the “mathematical formulation” of SDEs is “directly comparable” with the formulation of GANS.

I found this paper to be poorly premised. At the outset, the authors state “An SDE is a map from a noise distribution...to the solution of the SDE which is some other distribution on path space.” This statement is incorrect. First of all an SDE is not a map on measure space. It defines the evolution of sample paths of stochastic processes that induce measures. This suggests to me that the authors conflate the measures on path space with paths themselves. I’m also confused by the analogy between sampling SDEs and GANs — one might as well draw analogies with sampling Gaussian distributions. This is entirely confusing.

There are further fundamental issues that crop up throughout the paper. For instance, in Section 2.2, the authors state that $Y\stackrel{d}{\approx} Z$; but what do they mean by this? That  the finite dimensional distributions are approximately equal?

It appears that the point of the paper is that Wasserstein GANs can be applied to path measures induced by SDEs. This is not a novel insight, in my opinion.

---

> ### Author Response · Authors · 2020-11-13
> **Strong solutions to SDEs are maps between distributions**
>
> Thank you for your review.
>
> **Maps between distributions**
> The strong solution to an SDE may be defined as the unique map on Brownian noise satisfying the integral equation almost surely. The pushforward of this map on Brownian noise then gives the solution measure. See for example [Rogers & Williams, 2000, Chapter V, Definition 10.9].
>
> In this way the strong solution to an SDE is precisely a map between measures. There are indeed equivalent descriptions in terms of sample paths, but this is not the only description of an SDE.
>
> We have now expanded Section 2.1 to make this clear by laying it out explicitly. We would be happy to discuss this further as this seems to be a major factor towards the reviewer's score.
>
> **Other points**
> - The analogy between sampling SDEs and GANs is that one can (a) sample from an SDE or a GAN, and that (b) there is no natural or accessible notion of probability density for either an SDE or a GAN. If really desired the parameters of a Gaussian distribution could indeed be trained as a GAN, but it is much more practical to fit them by maximum likelihood -- an option unavailable to SDEs, which do not admit densities. This was discussed in Section 2.1, which we have now expanded greatly.
> - The notation $Y \overset{d}{\approx} Z$ is an informal one to describe the desired correspondence between model and data. This is described in the sentence following the use of this notation, which we have now expanded to ensure its clarity.
>
> **Summary**
> The review seems to be based upon a faulty understanding of SDEs. We would be happy to discuss this further as this seems to be a major factor towards the reviewer's score.

---

### Decision · Program_Chairs · 2021-01-07
**Final Decision**

**Decision:**

Reject

**Comment:**

The reviewers agree that this paper has some interesting ideas. However, they believe it needs more work before it is ready for publication, especially so with regards to presentation (SDEs as GANs) and the experiments (backpropagating through the solver rather than using the adjoint dynamics). These would significantly strengthen the paper, but would probably require another round of reviews.